# A Global Review on Short Peptides: Frontiers and Perspectives [note 1]

**DOI:** 10.3390/molecules26020430

**Published:** 2021-01-15

**Authors:** Vasso Apostolopoulos, Joanna Bojarska, Tsun-Thai Chai, Sherif Elnagdy, Krzysztof Kaczmarek, John Matsoukas, Roger New, Keykavous Parang, Octavio Paredes Lopez, Hamideh Parhiz, Conrad O. Perera, Monica Pickholz, Milan Remko, Michele Saviano, Mariusz Skwarczynski, Yefeng Tang, Wojciech M. Wolf, Taku Yoshiya, Janusz Zabrocki, Piotr Zielenkiewicz, Maha AlKhazindar, Vanessa Barriga, Konstantinos Kelaidonis, Elham Mousavinezhad Sarasia, Istvan Toth

**Affiliations:** 1Institute for Health and Sport, Victoria University, Melbourne, VIC 3030, Australia; vasso.apostolopoulos@vu.edu.au (V.A.); imats1953@gmail.com (J.M.); vanessa.barriga@live.vu.edu.au (V.B.); 2Institute of General and Ecological Chemistry, Faculty of Chemistry, Lodz University of Technology, Żeromskiego 116, 90-924 Lodz, Poland; 3Department of Chemical Science, Faculty of Science, Universiti Tunku Abdul Rahman, Kampar 31900, Malaysia; chaitt@utar.edu.my; 4Botany and Microbiology Department, Faculty of Science, Cairo University, Gamaa St., Giza 12613, Egypt; sh.elnagdy@gmail.com (S.E.); malkhazi@aucegypt.edu (M.A.); 5Institute of Organic Chemistry, Faculty of Chemistry, Lodz University of Technology, Żeromskiego 116, 90-924 Lodz, Poland; krzysztof.kaczmarek@p.lodz.pl (K.K.); janusz.zabrocki@p.lodz.pl (J.Z.); 6NewDrug, Patras Science Park, 26500 Patras, Greece; k.kelaidonis@gmail.com; 7Department of Physiology and Pharmacology, Cumming School of Medicine, University of Calgary, Calgary, AB T2N 4N1, Canada; 8Vaxcine (UK) Ltd., c/o London Bioscience Innovation Centre, London NW1 0NH, UK; r.new@vaxcine.co.uk; 9Faculty of Science & Technology, Middlesex University, The Burroughs, London NW4 4BT, UK; EM1081@live.mdx.ac.uk; 10Center for Targeted Drug Delivery, Department of Biomedical and Pharmaceutical Sciences, Chapman University School of Pharmacy, Harry and Diane Rinker Health Science Campus, Irvine, CA 92618, USA; parang@chapman.edu; 11Centro de Investigación y de Estudios Avanzados del IPN, Departamento de Biotecnología y Bioquímica, Irapuato 36824, Guanajuato, Mexico; octavio.paredes@cinvestav.mx; 12Infectious Disease Division, Department of Medicine, Perelman School of Medicine, University of Pennsylvania, Philadelphia, PA 19104-6073, USA; hamideh.parhiz@pennmedicine.upenn.edu; 13School of Chemical Sciences, The University of Auckland, Private Bag 92019, Auckland 1142, New Zealand; c.perera@auckland.ac.nz; 14Departamento de Física, Facultad de Ciencias Exactas y Naturales, Universidad de Buenos Aires, Buenos Aires 1428, Argentina; monicapickholz2@gmail.com; 15Instituto de Física de Buenos Aires (IFIBA, UBA-CONICET), Argentina, Buenos Aires 1428, Argentina; 16Remedika, Luzna 9, 85104 Bratislava, Slovakia; milan.remko@gmail.com; 17Institute of Crystallography (CNR), Via Amendola 122/o, 70126 Bari, Italy; michele.saviano@cnr.it; 18School of Chemistry & Molecular Biosciences, The University of Queensland, St Lucia, QLD 4072, Australia; m.skwarczynski@uq.edu.au (M.S.); i.toth@uq.edu.au (I.T.); 19Key Laboratory of Bioorganic Phosphorus Chemistry & Chemical Biology (MOE), School of Pharma Ceutical Sciences, Tsinghua University, Beijing 100084, China; yefengtang@tsinghua.edu.cn; 20Peptide Institute, Inc., Osaka 567-0085, Japan; t.yoshiya@peptide.co.jp; 21Institute of Biochemistry and Biophysics, Polish Academy of Sciences, Pawinskiego 5a, 02-106 Warsaw, Poland; piotr@ibb.waw.pl; 22Department of Systems Biology, Institute of Experimental Plant Biology and Biotechnology, University of Warsaw, Miecznikowa 1, 02-096 Warsaw, Poland; 23Institute for Molecular Bioscience, The University of Queensland, St Lucia, QLD 4072, Australia; 24School of Pharmacy, The University of Queensland, Woolloongabba, QLD 4102, Australia

**Keywords:** short peptides, constrained amino acids and peptide (bio)mimetics, drug design and drug/gene delivery, vaccines, aptamers, cell-penetrating peptides, synthesis, SARS-COV-2, cancer, bilayer interactions, altered peptide ligands, ant/super/agonists, diketopiperazine, cosmeceuticals

## Abstract

Peptides are fragments of proteins that carry out biological functions. They act as signaling entities via all domains of life and interfere with protein-protein interactions, which are indispensable in bio-processes. Short peptides include fundamental molecular information for a prelude to the symphony of life. They have aroused considerable interest due to their unique features and great promise in innovative bio-therapies. This work focusing on the current state-of-the-art short peptide-based therapeutical developments is the first global review written by researchers from all continents, as a celebration of 100 years of peptide therapeutics since the commencement of insulin therapy in the 1920s. Peptide “drugs” initially played only the role of hormone analogs to balance disorders. Nowadays, they achieve numerous biomedical tasks, can cross membranes, or reach intracellular targets. The role of peptides in bio-processes can hardly be mimicked by other chemical substances. The article is divided into independent sections, which are related to either the progress in short peptide-based theranostics or the problems posing challenge to bio-medicine. In particular, the SWOT analysis of short peptides, their relevance in therapies of diverse diseases, improvements in (bio)synthesis platforms, advanced nano-supramolecular technologies, aptamers, altered peptide ligands and in silico methodologies to overcome peptide limitations, modern smart bio-functional materials, vaccines, and drug/gene-targeted delivery systems are discussed.

1. Introduction

2. Brief History

3. Short Peptides: Definition

4. Frontiers and Prospects of Short Peptides

4.1. Advantages vs. Disadvantages: SWOT Analysis

4.2. To Overcome Shortcomings of Peptides: Mission (Im)possible?

4.2.1. Constrained Amino-Acids as a Molecular “Meccano”

4.2.2. Cyclic Peptides and Mimetics

4.2.3. Ultra-Short Peptides: Less Is More

4.2.4. Nanoengineering & a Supramolecular Approach

5. Synthesis

5.1. Advances in the Synthesis of Short Peptides and Modified Amino Acids

5.2. Short Difficult Peptide Synthesis

6. In Silico Studies

6.1. Geometry Optimization, Conformational Analysis

6.2. Modelling of Short Peptides

6.3. Peptide Interactions with Lipid Bilayers using Molecular Dynamics Simulations

7. Peptide-Based Therapies

7.1. Monocyclic, Bicyclic and Tricyclic Cell-Penetrating Peptides as Molecular Transporters

7.2. Short Peptides in Gene Delivery

7.3. Taking Peptide Aptamers to a New Level

7.4. Peptide-Based Vaccines

7.5. The Role of Short Peptides in Neurodegenerative Therapy

7.6. Immune Modulation Using Altered Peptide Ligands in Autoimmune Diseases

7.7. Relevance of Short Peptides in Stem Cell Research

7.8. Short Peptide-Based Anti-Viral Agents against SARS-CoV-2

7.9. Antimicrobial Lactoferrin-Based Peptides as Anti-COVID-19

7.10. Peptides from Digestion of Proteins

7.11. Nutraceuticals

7.12. Marine Peptides

7.13. Peptide-Based Cosmeceuticals

## 1. Introduction

Recently, short peptides have attracted increasing attention in biology, chemistry, and medicine due to their specific features. They are appreciated as novel and more efficient therapeutical agents with reduced side effects. Their structural diversity combined with the conformational flexibility is used to control interactions with particular receptor sites. Peptides display high selectivity due to specific interactions with their targets. Moreover, the number of short peptides involved in important biological processes is steadily growing by far exceeding that resulting from the traditional mimetic approach. Unfortunately, peptides also have profound medical limitations, namely the development of oral peptide-based therapeuticals that modulate cellular processes via high affinity binding is like a search for the Holy Grail [1].

This critical review is written by a broad, multidisciplinary group of leading scientists, experts in the field from academia and pharmacy from all continents of the world providing a priceless global point of view on short peptides towards biomedical innovations. It is compiled as a holistic story from very simple bio-molecules to next-generation advanced theranostics in diverse, multidirectional scenarios. More specifically, the advantages vs. disadvantages of short peptides, their relevance in therapies of a wide range of diseases, improvements in (bio)synthesis platforms, advanced nano-supramolecular technologies, aptamers, altered peptide ligands and in silico methodologies to overcome peptide limitations, modern smart bio-functional materials, vaccines, and drug/gene-targeted delivery systems are considered. It gains unequaled insight into the world of functional biologically active peptides either accessible from nature’s repertoire or synthetic species. Short peptides, as fragments of proteins, have become intriguing agents of almost unlimited possibilities, which are awaiting further exploitation in the near future. We profoundly believe that these simple bio-molecules will open up whole new vistas, offering promising solutions in shaping the future novel bio-medicine.

## 2. Brief History

In 1902, two distinguished German chemists, Hermann Emil Fischer and Franz Hofmeister, proposed that proteins are constituted by amino acids linked by bonds between the amino group of the proceeding amino acid and the carboxyl group of the following residue [2]. However, proteins were initially characterized by the Dutch chemist Gerardus Johannes Mulder, but their name was coined out by the Swedish chemist Jöns Jacob Berzelius, in 1838 [3,4]. The term “protein” is derived from the “proteios” (“primary”) i.e., representing the first position in living organisms [4,5,6]. Nevertheless, proteins do not exist without peptides. A name “peptide” comes from “peptós” (in Greek “digested, digestible”) and reflects the fact that peptides are generated by the proteolytic cleavage reaction. The first peptides and amino acids were discovered at the beginning of 19th century [7,8]. The first amino acid, asparagine, was isolated from asparagus by French chemists Louis-Nicolas Vauquelin and Pierre Jean Robiquet in 1806 [9,10]. Their chemical category was recognized by the French Charles Adolphe Wurtz, in 1865, but the expression “amino acid” was used for the first time in 1894, in German as *Aminosäure* [11,12]. Interestingly, the first peptide, benzoylglycylglycine, was synthesized by the German chemist Theodor Curtius, in 1881 [13]. However, a more efficient synthesis was described by Fischer and the French chemist Ernest Fourneau in 1901 [14,15]. In consequence, Fisher is known as the “father” of peptide chemistry [16].

Peptides exist in all terrestrial living organisms and are indivisibly related to the origin of life [17]. Cooperative interactions among peptides and other molecules (amino acids, proteins, nucleic acids, lipids) were the driving forces at all stages of chemical evolution [18]. Nowadays, a chemical peptide synthetic biology approach facilitates theories on the creation of life, in particular in the eyes of scientists who believe that historically chemistry proceeds biology [19,20,21].

## 3. Short Peptides: Definition

In general, a peptide consists of at least two amino acids. An oligopeptide is a short chain of amino acids (“a few”). A polypeptide is a long chain of amino acids (“many”). Protein contains at least one polypeptide chain folded into correct shapes. There is no strict boundary between a peptide and a protein or an oligopeptide and a polypeptide other than the “size”. As stated in the International Union of Pure and Applied Chemistry (IUPAC), oligopeptides consist of fewer than about 10–20 amino acids, while polypeptides have more than 20 residues [22]. According to the biological dictionary, oligopeptides comprise about 2–40 amino acids, while the medical definition indicates a fragment of protein consisting of fewer than 25 amino acids. On the other hand, proteins, according to IUPAC can be polypeptides consisting of more than about 50 mers, but there are great differences regarding the term protein. In the Britannica encyclopedia, we can read: “peptide chains longer than a few dozen amino acids are called proteins” [23]. Typical proteins contain over 100 amino acids [5,24]. The smallest natural mini-protein is crambin, consisting of 46 amino acids [25], while the largest protein is titin with 38,138 amino acid residues [26]. Hence, the determination of “short peptide” is problematic. It depends on the reference point. The strict definition has not been given so far. Short peptides have features of oligomers rather than polymers [5], but there is no clear consensus among scientists. In the literature, we can find contradictory information, with fewer than 30 [27,28,29] or 50 [30,31,32,33], up to 100 residues [34]. On the other hand, ultra-short peptides were precisely defined as peptides consisting of up to seven amino acids [35,36,37,38].

In view of the above, we can conclude that oligopeptides are always only peptides, while polypeptides can be proteins as well. Consequently, short peptides should not include more than 45 amino acids.

## 4. Frontiers and Prospects of Short Peptides

### 4.1. Advantages vs. Disadvantages: SWOT Analysis

Peptides as a unique class of bio-molecules have filled the therapeutic niche due to their specific biochemical and therapeutic features. They explore the “middle space” between small chemical molecules and biologics because of their molecular weight. They have the intermediate nature extending “beyond size”, combining the advantages of both small molecular drugs (e.g., better permeability) and therapeutic proteins (selectivity, target potency) and exluding their disadvantages, such as adverse side effects, drug-drug interactions, and membrane impermeability, respectively.

Short peptides have evolved as a very promising scaffold for diverse applications either in diagnosis or therapies. The current status of their strengths, weaknesses, opportunities and threats (SWOT analysis) [39] is briefly discussed (Table 1).

First of all, short peptides have numerous advantages in comparison with their larger analogues. In particular, cost-effective synthesis both on a small- and large-scale, wide chemical diversity, easy modification, high bio-activity, absorbability, accessibility, tunable functionalization, high selectivity and specificity, biodegradability and biocompatibility, high safety, low toxicity (due to their safe metabolites-amino acids, the limited possibility for accumulation in the body), or low immunogenicity should be emphasised [40]. Peptides have diverse bio-functionalities of their components (amino acids) and good biomolecular recognition [34,41]. As a consequence, they have high binding affinity for a wide range of specific targets.

On the other hand, short peptides have limitations, such as high conformational flexibility (can result inter alia in the lack of receptor selectivity) or problems in permeability of greater peptides via physicological barriers (due to the strong interactions of peptide backbone with water molecules) [42]. Moreover, there are other important factors, e.g., short half-life in vivo (due to the susceptibility to rapid digestion by protolytic enzymes in the gastrointestinal tract and serum, proteases/peptidases) and fast clearance from the circulation (first-pass metabolism) by the liver and kidneys (lasting from minutes to hours after administration). In spite of approvement of over 60 peptide drugs, nearly none can be orally administrated [43]. Market placement of effective peptides as oral medications is still the “Holy Grail”. Furthermore, the risk of immunogenic effects is the main threat of peptide therapies [42].

### 4.2. To Overcome Shortcomings of Peptides: Mission (Im)possible?

There are different approaches and strategies to overcome peptide limitations and enhance their bio-clinical applications [44]. First and foremost, structural modifications can lead to the improvement of physicochemical properties. Simple modifications result in greater general stability. It can be additionally improved by “double-bridged peptides” when peptides are cyclized via two chemical bridges. It reduces peptide backbone flexibility and in consequence, leads to limited availability for enzymes. Furthermore, variations in the sequence lenght and side chains, peptide backbone modification, C-terminal amidation, N-terminal acetylation, addition of stabilizing (sugars, salts, heparine) and chelating agents, e.g., ethylenediaminetetraacetic acid (EDTA), conjugations with large biocompatible polymers, such as polyethylene glycol (PEG), or fatty acids can be applied. In this way, we can stabilize peptides in their bioactive conformation, increase efficiency, hydrodynamic volume, and reduce renal clearanced and show greater membrane permeability and target selectivity [33,41,42,43,44,45]. Furthermore, the conjugation to cell-penetrating peptides (or organelle-targeting sequences) increases cellular membrane crossing and allows accessing intracellular targets by peptide-drugs or acts as gene delivery vectors revealing great potential for clinical use as theranostics leading to better drug bioavailability and therapeutic efficiency [46,47]. The conjugations of short peptides with non-peptidic motifs enhance bioactivities: A promising strategy for the discovery of new drugs (improve peptide delivery and cellular uptake) [48]. Synthetic short peptides as accurate copies of protein parts are ideal tools for imitation of protein sites [47]. Peptoids, which are based on native peptides, can lead to the improved pharmacokinetic profile [49,50]. Novel methods such as phage display can be used to develop short peptides, which can survive proteolytic degradation in the gastrointestinal tract and can be used as therapeutical agents with high affinity in inhibition of the coagulation Factor XIa or as antagonists for the interleukin-23 receptor in the chronic inflammatory Crohn‘s disease, ulcerative colitis [51]. They may be a milestone towards engineering oral peptide drugs in the treatment of diseases affecting billions of people worldwide [52,53].

#### 4.2.1. Constrained Amino-Acids as a Molecular “Meccano”

Intensive efforts have been made to develop short peptides or peptidomimetics that display more favourable pharmacological properties than their prototypes [54]. Most of the research carried out in the field concern the preparation of analogues with different chemical structure and possibly modified conformational preferences, responsible for inducing changes in the biological activity. Structural changes can be obtained in a peptide by selectively substituting along the sequence specific residues with other residues or by substituting certain residues of the sequence with non-coded α-amino acid residues. Appropriate constrained non-coded α-amino acid residues are of particular interest as “building blocks” for the preparation of analogs, since their inclusion in a peptide sequence could maintain the pharmacological properties of the native peptide and possibly enhance resistance to biodegradation with improved bioavailability and pharmacokinetics. Several solid-state studies have been carried out to define the conformational preferences in solution and in the solid-state of specific classes of non-coded α,α -amino acids, for example, the symmetrical and unsymmetrical α,α-disubstituted glycines (α,α-dialkylated amino acid residues) (Figure 1) [55,56,57,58,59,60,61,62,63,64,65,66]. The structural preferences of peptides containing non coded amino acid residue are unique with significant constraints of their conformational freedom. This point is particularly important for the use of these residues and their analogues as scaffolding units in the de novo design of protein and enzyme mimetics and, also, as templates for molecular and chiral recognition studies. More in general with this knowledge we are able to rationally design new peptides relevant to pharmacology and medicinal chemistry, which might mimic biological processes by enhancing or in general modulating their effects. The peptide pharmaceutical targets of these studies have been among others hormones, enzymes, transport systems, antibiotics, sweeteners, etc. [67,68,69,70,71,72]. Another important application of constrained amino acid is in the peptide self-assembly. This process governs the organization of proteins, controlling their folding kinetics and preserving their structural stability and bioactivity. In this connection, model oligopeptides containing α,α-disubstituted glycines can give important insights into the molecular mechanisms and elementary forces driving the formation of supramolecular structures with potential application in tissue engineering [73,74].

#### 4.2.2. Cyclic Peptides and Mimetics

Cyclic peptides constitute a class of compounds that were used in the treatment of certain diseases. Examples of such well known cyclic peptides are insulin, penicillin, cyclosporin, and gramicidin S. Cyclic peptides, compared to linear peptides, have been proved to show greater potential as therapeutic agents due to their increased chemical and enzymatic stability, receptor selectively and improved pharmacodynamic properties. In our peptide research, cyclization of peptides is a key step towards non peptide mimetics which is the final target [75]. Our research group was the first, worldwide, to synthesize cyclic analogues of important peptides such as angiotensin II (implicated in hypertension), myelin epitope peptides (implicated in multiple sclerosis), gonadotropin releasing hormone (implicated in infertility and cancer), and thrombin receptor activating peptides (implicated in angiogenesis and cancer) [76,77,78,79,80,81,82,83,84]. Another way of transforming peptides to peptide mimetics is by conjugating peptides to sugars like mannan, used as antigen carriers in cancer and in multiple sclerosis research [85]. The octapeptide hormone angiotensin II is one of the best studied peptides with the aim to design and synthesize non peptide mimetics for oral administration [75,86,87]. To achieve this target, cyclizations at different positions within the peptide molecule was a useful strategy to define the active conformation [78,79,80,81]. These studies on angiotensin II led to the discovery of sarmesin, a type II angiotensin II antagonist and the breakthrough non-peptide mimetic losartan, the first in a series of sartans marketed as a new generation of anti-hypertensive drugs in 1990s [78,79,88,89]. These studies led also to the ring cluster conformation of angiotensin II and the charge relay system hypothesis confirmed by fluorescence studies [90]. Synthesis of cyclic peptides, in our studies, was pursued as an intermediate step towards constructing non peptide mimetics which as drugs have the merit to be administered orally. The limited stability of peptides, due to hydrolysis of amide bonds, severely restricts their medical and industrial application. Therefore, the engineering of stable peptide moieties, which are the cyclic counterparts and non-peptide mimetics is of outmost importance. Furthermore, cyclizations were a way to define and lock the active conformation of the peptide. Structure–activity studies have shown the importance of the three aromatic amino acids Tyr, His, Phe, and the C-terminal carboxylate for activity. In order for cyclic analogues to retain the activity of the linear peptide, cyclization should occur at residue positions that are the least important for activity with retention of the bioactive conformation [78,79,80,81]. The conformation of peptides is deduced from modern Nuclear Magnetic Resonance techniques, such as two-dimensional (2D) NMR ROESY, NOESY, COSY, and TOCSY in lipophilic environments. Based on losartan, sarmesin, and our ring cluster and charge relay system conformation, we designed and synethsized angiotensin II receptor blockers by rotation of the alkyl chain on the imidazole ring. This rotation resulted in losartan V8 and BV6 derivatives of similar activity with losartan [91,92]. The perspectives in the use of angiotensin receptor blockers (ARB) are huge. ARB and angiotensin-converting enzyme (ACE)1 inhibitors were recently found to protect hypertensive patients infected from Severe Acute Respiratory Syndrome Coronavirus (SARS-Cov-2) [93]. The renin-angiotensin system (RAS) inhibitors reduce the toxic angiotensin II and increase antagonist heptapeptides alamandine and aspamandine, which counterbalance angiotensin II and maintain homeostasis and vasodilation [94] (Figure 2).

#### 4.2.3. Ultra-Short Peptides: Less Is More

Very short peptides have outstanding attributes, such as much easier and economic synthesis, higher mechanical stability, good tissue penetration, and less immunogenicity [42]. Overall, the costs of production increase with the lenghth of the amino-acids chain. In addition, a complex control and characterization are simpler. Furthermore, very short peptides have better tunability, bio- and cytocompatibility (non-hemoliticity and non-cytotoxicity) [95], biodegradability, and non-immunogenicity, in comparison with their longer analogs. They can support the growth of diverse cells and their differentiation [96]. As a result, structural optimization is easier. Ultra short peptides are more amenable to oral delivery. They contain necassary molecular information on spontaneous self-assemble into the ordered nanostructures [97]. Fluorenylmethoxycarbonyl(Fmoc)-diphenylalanine is a good example of highly ordered peptide (hydrogel) with antimicrobial activity [98,99] leading to the acceleration of wound healing [100]. Ultra-short nanoparticles can overcome the drug problem related to low half-life. As an example, encapsulation of curcumin in the form of nanocarrier can be used in a “controllable release” way to repair brain tissues as a promising drug in neurodegenerative diseases [101]. Moreover, ultra-short peptides are suitable for many bioapplications, innovative nano-theranostics (either therapeutic or diagnostic), especially in cancer cell growth inhibition, and advanced smart system formulations [102,103,104,105,106]. The latter include oral administration which increases drug efficacy and safety.

The simplest cyclo-peptides, also known as 2,5- (and 2,3- or 2,6-) diketopiperazines (DKP), piperazine-2,5-diones, 2,5-dioxopiperazines, and dipeptide anhydrides, are another important issue. They have amazing advantages in drug discovery due to their extra features, inter alia superior rigidity, three-dimensionality, higher cell permeability [107], and diverse bio-activities: Anticancer, antiviral, antioxidant, in neurodegeneration prevention, quorum sensing, cell-cell signaling, as drug delivery systems (e.g., in connection with cell-penetrating peptides) and so on [108,109,110]. DKPs are widespread in nature. They occur either in the marine and terrestrial environment, in microorganisms or high species, or in food and drugs [111,112]. In the latter case, DKPs are inner cyclization products of bio-active substances, e.g., ACE inhibitors [113,114,115,116,117,118,119,120,121,122]. Noteworthily, they have similar pharmacological profile as corresponding drug molecules. They were found in the central nervous system, gastrointestinal tract, and blood [123]. Interestingly, they played a key role in the origin of life [112,124]. The growing interest in these simplest natural cyclo-peptides is noticeable due to their huge potential in future therapies. 2,5-DKPs are observed in almost 50 bio-complexes deposited in the Research Collaboratory for Structural Bioinformatics Protein Data Bank (RCSB PDB) [125] (Figure 3).

Generally speaking, short peptides are gaining more and more popularity. Exploration of their new modifications seems promising for new advanced research and will be the key to innovative smart bio-medical applications [126].

#### 4.2.4. Nanoengineering and a Supramolecular Approach

Supramolecular chemistry (‘‘chemistry beyond the molecule’’) is a bottom-up approach to the formation of well-ordered structures in the nano- (and the micro-) scale. Their adaptable, controllable, self-healing, and bio-physico-chemical stimuli-responsive properties induced via non-covalent interactions (electrostatic, hydrogen bonding, π–π stacking, van der Waals or hydrophobic) are highly appreciated [123,127,128,129,130,131,132]. They have the dynamic nature and provide a firm basis for the structure and functioning of living systems. Notably, bio-systems are controlled by a plethora of supramolecular interactions. The formation of deoxyribonucleic acid (DNA) double helix via H-bonding interactions between the nucleo-bases or the folding of proteins into tertiary and quaternary structures is a good example of supramolecular assemblies. The self-assembly is a spontaneous, reversible, and ubiquitous process [98,127]. Self-assembling peptides are favorable platforms for the development of next-generation smart therapies. Minor structural changes can facilitate the generation of new assemblies. Modified peptides with aromatic amino acids or functionalized side chains (e.g., Fmoc) promote additional stacking interactions (Figure 4), which are helpful in the self-assembly process. Peptide-based bio-functional supramolecular materials (nanomedicines, hydrogels, drug delivery vehicles, gene or drug carriers, biomimetic-cell culture scaffolds, tissue-engineering systems, biosensors, emulsifiers, peptidomimetic antibiotics, bioimaging nanoprobes, three-dimensional (3D) bioprinting inks, vaccine adjuvants) have low toxicity and high biocompatibility and are useful in various applications, like drug delivery, tissue engineering, immunology, cancer therapy, and stem cell culture [38,95,101,131,132,133,134,135,136]. Supramolecular nanotherapeutics have better stability and efficacy, which helps to overcome problems related to peptide poor biostability and short plasma half-life. Nanoparticles’ conjungtion with inter alia tumor-homing peptides is an attractive avenue for tumor-targeted therapy [136,137]. The advances in the synthesis of supra-molecular assemblies prompt the development of theranostics (which possess specific smart features, such as programmability, multifunctionality, sensitivity, precise selectivity, biosafety) and are promising agents for personalized, smart medicine [126,138,139].

## 5. Synthesis

### 5.1. Advances in the Synthesis of Short Peptides and Modified Amino Acids

The synthesis of modified amino acids and short peptides is based on reactions of functional groups. An example of modified amino acids at α-carbon atom could be α-hydroxymethylvaline [128], while 4-aminopyroglutamic acid can serve as cyclized dipeptide unit [140].

The typical chemical peptide synthesis requires the condensation reaction of the carboxyl group of one amino acid with the amino group of another. The only two approaches to a peptide synthesis are the synthesis in solution and the solid-phase synthesis (SPPS). In each of them, other steps besides peptide bond formation and/or deprotection, like introduction of main- and side-chain modifications, are available. The liquid-phase approach is now used mostly for the synthesis of short peptides up to tetrapeptide. During solid-phase peptide synthesis, each peptide is anchored at the C-terminus or side chain functional group to an insoluble/soluble polymer. In both cases, a single N-protected amino acid unit is coupled to the free N-terminal amino group of growing peptides. After deprotection, which reveals a new amino group, another amino acid may be attached. Once synthesis is complete, the desired peptide is cleaved from the resin. Usually, this cleavage step is performed with acids of varying strength. Any functionalized polymer could be used as a solid support, like for instance styrene cross-linked with 1–2% divinylbenzene [141], which is a popular carrier resin in SPPS. Other common gel-type supports include polyacrylamide and polyethylene glycol (PEG).

In solution peptide synthesis, each step requires precise product purification using gel chromatography or crystallization. In contrast, in SPPS purification is performed as a simple washing of the peptide attached to the polymer. This allows to design peptide synthesizers and automate the synthetic procedure.

The most commonly used peptide coupling reagents can be divided into two classes: Older reagents—carbodiimides and newer—salts of 1-hydroxy-benzotriazole (HOBt) and its derivatives. A danger of racemization during carbodiimide activation can be circumvented through the addition of ‘racemization suppressing’ additives such as triazoles (HOBt) and its derivatives (for example 6-Cl-HOBt) or 1-hydroxy-7-aza-benzotriazole (HOAt). A more recently developed additive for carbodiimide coupling with comparable coupling efficiency to HOAt is ethyl cyanohydroxyiminoacetate (Oxyma) [142]. As newer and commonly used reagents, there are compounds contained in the structure HOBt (HBTU/TBTU/PyBOP), HOAt (HATU), 6-ClHOBt (HCTU). Their iminium, uronium, or phosphonium salt of a non-nucleophilic anion (tetrafluoroborate or hexafluorophosphate) act as coupling reagent with the ability of lowering racemization process [143].

The amino acids have to be orthogonally protected during any chemical reaction, because of the presence of two or more reactive centers. In typical peptide synthesis, orthogonal but permanent protection is necessary for the amino acids’side-chain functionalities. Another class of protecting group is required for temporary protection of the α-amino group. The temporary protecting group is being cleaved repeatedly while the permanent protecting groups have to withstand N-deprotection conditions. When the synthesis is complete, they can be removed from the final peptide separately or simultaneously. Two principle orthogonal protecting group schemes are utilized in solution and/or solid phase synthesis: So-called Boc/Bzl (tert-butyloxycarbonyl/benzyl) and Fmoc/*t-*Bu (fluorenylmethylenoxycarbonyl/tert-butyl) approaches [144]. The Boc/Bzl-strategy requires side chain protecting groups, which should be stable during repetitive trifluoroacetic acid treatment. SPPS can be automated far more conveniently for Fmoc/tert-butyl (tBu) than for Boc/Bzl strategy. The advantage of Fmoc is that it is cleaved under very mildly basic conditions, but it remains stable under acidic conditions. This allows the use of mild acid-labile protecting groups, such as Boc and tBu groups, to be used on the side-chains of amino acid residues of the target peptide. Among other protecting groups recommended for special cases are for example allyloxycarbonyl (Alloc) for amino group and allyl ester for carboxyl group. There is always the possibility for their selective removal in the presence of Fmoc/tBu protections.

Solution phase synthesis as well as polymer supported synthesis can be utilized for the modification of amino acids and/or amide bonds. Nowadays, among popular modifications are all kinds of cyclization. Peptides can be cyclized on a solid support and/or in a solution with any coupling reagent. The disadvantage of the solution phase cyclization is a substrate high dilution necessity to limit the possible reactions to intramolecular one. The solid-phase synthesis of head-to-tail cyclic peptides is not limited to attachment through Asp, Glu, or Lys side chains. For example, cysteine has very reactive sulfhydryl group, which can be utilized as an anchoring point.

### 5.2. Short Difficult Peptide Synthesis

Although many peptides are routinely synthesized by SPPS, a certain kind of peptides is still difficult to be prepared. Such peptides are called “difficult sequence-containing peptides” [145]; even pentapeptides such as Ac-Val-Val-Ser-Val-Val-NH_2_ are known as an example. During the chain elongation, intra-/inter-molecular hydrophobic interactions and/or hydrogen bondings cause aggregation of protected peptides on the resin to induce incomplete coupling and deprotection. Furthermore, after final deprotection, hydrophobic peptides hamper HPLC purifications using H_2_O-MeCN system. Modifying the main chain amide bond by e.g., pseudoproline (ΨPro) method [146] or *O-*acyl isopeptide method [147,148,149] often solves such problems. In these methods, main chain amide causing the undesired secondary structures is protected or modified to improve SPPS efficiency. For example, diabetes mellitus-related amylin is difficult to be prepared by the traditional Fmoc SPPS but was synthesized successfully with the aid of ΨPro structure [150,151]. In the ΨPro method (Figure 5), side chain of Ser/Thr/Cys in the difficult peptide is protected as Pro-mimicking “ΨPro” during SPPS and final trifluoroacetic acid (TFA) treatment liberates native Ser/Thr/Cys. Because N-alkylated amide cannot become a hydrogen bond donor and N-alkylation of the amide bond affects the cis/trans ratio of the amide bond, this modification drastically changes secondary structure of the growing protected peptide on the resin [152,153]. The ΨPro method successfully applied to the synthesis of short cyclic peptides as well [154,155]. As described above, during TFA-mediated final deprotection reaction, the native peptide is released. This tracelessness is a strong point but sometimes becomes a weak point; if the native peptide is hydrophobic, the following high-performance liquid chromatography (HPLC) purification would be difficult. To overcome it, the *O-*acyl isopeptide method was developed (Figure 6). In this method, the target peptide is synthesized in a form of an *O*-acyl isopeptide, which contains an *O*-acyl isopeptide bond instead of the native N-acyl peptide bond at a hydroxy group-containing amino acid residue, e.g., Ser or Thr. Namely, in this method, the main chain α-amide is changed to a β-ester bond, which is not a hydrogen bond donor and has no cis/trans-causing rotation barrier. Thus, incorporation of the isopeptide structure drastically changes the secondary structure to increase the efficacy of peptide preparation during chain elongation in a similar manner to the ΨPro method. Meanwhile, in contrast to the ΨPro method, such an *O*-acyl isopeptide is stable under acidic conditions or as a powder (e.g., a lyophilized TFA salt) and the target peptide can be quantitatively obtained by a quick and one-way *O*-to-N intramolecular acyl migration reaction under neutral conditions from the corresponding *O*-acyl isopeptide. Thus, hydrophobic peptides can be efficiently purified by HPLC (high performance liquid chromatography) at the *O*-acyl isopeptide stage [156]. The *O*-acyl isopeptide can be readily synthesized with *O*-acyl isodipeptide units [157,158,159]. Isodipeptide units have enabled routine application of the *O*-acyl isopeptide method by omitting the often-difficult esterification reaction on a resin. So far, many difficult sequence-containing peptides were synthesized by this method. Firstly, this method successfully applied to the synthesis of difficult pentapeptides (Ac-Val-Val-Ser-Val-Val-NH_2_ [147] and Ac-Val-Val-Thr-Val-Val-NH_2_ [157]). Later, Cys was used instead of Ser/Thr as the *S*-acyl isopeptide method to prepare Ac-Val-Val-Cys-Val-Val-NH_2_ [160]. Since then, e.g., Alzheimer’s disease-related amyloid β peptide 1-42 (Aβ42) [40,161], amylin [162], vaccine peptide [163], insulin derivatives [164,165,166,167], and collagen peptide [168] are efficiently prepared by the *O*-acyl isopeptide method. Especially, in case of highly aggregative Aβ42, isoAβ42 was confirmed being monomeric without any pretreatment [169]. The *O*-acyl isopeptide method is also applied to the peptide cyclization and segment condensation [170,171,172,173,174]. As described in this section, the ΨPro method and the *O*-acyl isopeptide method will assist Fmoc SPPS of difficult peptides in future.

## 6. In Silico Studies

### 6.1. Geometry Optimization, Conformational Analysis

In peptides and proteins, individual amino acids are linked together through peptide bonds. The peptide bond is planar with a significant double bond character, which prevents rotation. Thus, bond rotation in the short peptide backbone is only possible for the bonds on either side of the peptide bond. Molecular modeling tools allows constructing a 3D structure of a short peptide on a computer [175]. Once the structure is built, the process of geometry optimization should be carried out. The main objective of this process is to find the lowest energy conformation. Conformations with the lowest energy correspond to the most stable molecules. However, methods for finding minima normally can find local minima only. Therefore, to find the global minimum you must check various possibilities of geometrical arrangement to see which one is the global minimum on the multidimensional potential energy surface. One of the possibilities for scientists to do this is to create a potential energy surface (PES). A short peptide consisting of to 50 amino acids has many degrees of freedom and consequently a complicate PES. The geometry optimization of short peptide can be carried out using both quantum chemical and/or molecular mechanic methods. Over the past decades, novel quantum chemical methods have been developed that are capable of treating peptides at a high level of electron correlation [176,177,178,179]. Benchmark quantum chemical computations were also used to determine the accuracy of force fields in ranking compact, low-energy peptide structures [176,178,180]. However, achieving a reasonable description of a potential energy surface for thermal body temperature (i.e., RT(310K) = 0.616 kcal/mole) exceeds the limits of modern force fields [181,182]. The structure obtained by energy minimization, however, is not necessarily the most stable conformation. Energy minimization will stop when it reaches the first stable conformation, and it usually encounters a local energy minimum. The minimization program has no manner to identify that there is a more stable conformation and converge the optimization process to a global minimum. To identify the most stable conformation of peptide it is necessary to generate different conformations and compare their energies. There are two methods of doing this, using molecular dynamics and/or stepwise rotation of bonds (conformational searching). Conformational searching is feasible for small oligopeptides only. When the number of rotatable bonds exceeds 5, the entire conformational space of a molecule can become extremely large. In recent years, the methods of molecular dynamics were increasingly used to gain insight into the structure/function relationships in short peptides and proteins [183,184,185,186,187]. One of the key questions to be answered when checking the applicability of molecular dynamic simulations for peptides and/or proteins is the extent to which the simulations appropriately sample the conformational space of these molecules. If a given property is poorly sampled over the molecular dynamics (MD) simulations, the results obtained have limited usefulness [184]. To improve the sampling efficiency, new techniques were developed [184,185,187]. All-atom molecular dynamic simulations can predict structures of peptides and other peptide foldamers with accuracy of experiments [187,188]. Thus, MD simulations are a useful tool for prediction and/or reproduction of experimental 3D structures of small proteins.

### 6.2. Modelling of Short Peptides

The growing interest in peptide research results from recent successes in designing peptide sequences of therapeutic value. At present, more than 60 peptide drugs have been approved for use, with hundreds being in clinical trials. A number of new methodologies, both experimental and in silico, have been developed to design active peptide sequences and analyze peptide-protein interactions. The above-mentioned issues are discussed in a recent comprehensive review [44]. Here, we concentrate on the problems encountered both on modelling as well as practical sides. From the theoretical point of view, they result mainly from the intermediate size of peptides—neither relatively exact calculations for small molecules, nor very approximate but often successful methodologies used for protein-protein recognition, work properly for them. This is because the size and flexibility of peptides are prohibitive for exact calculations on the one hand, excluding the error-compensation mechanism, which we benefit from in addressing protein-protein recognition problems on the other. The practical biological problem results from many roles short peptides play in the living cells, many of which are not understood properly until now. It may happen that the designed peptide is either: The (folding) inhibitor of the essential protein or protein-protein (interaction) or protein function inhibitor also interacting with a different protein that it was designed to bind to; executes a biological function for reason which is not meant or understood. There are numerous examples of these effects, e.g., the peptide corresponding to the (83–93) segment of human immunodeficiency virus (HIV) protease interferes with the formation of the (post-critical) folding nucleus and inhibits folding of the protein [189], and native interface peptide fragments can be used as proton-pump inhibitors inhibitors (PPI). It is reasonable to assume that protein degradation products can affect PPIs in a similar manner [190], whereby short peptides can affect cellular processes in a way beyond explanation at the current state at knowledge e.g. there are more than 1700 peptides (sometimes fragments of angiotensinogen) known to lower arterial hypertension [191]—the mechanism of their action is unknown. In addition, our limited experience in targeting PPIs with peptides in bacterial cells for therapeutic purposes [189] shows that a significant fraction (~20%) of designed peptides is deadly for unknown reason.

In conclusion, interactions of peptides (both native and designed) are of great importance for the metabolic processes of the living cells. Their full understanding requires further progress on both modelling and thermodynamic aspects of their interactions as well as understanding the peptide content of cells coming from both synthetic and protein degradation corners.

### 6.3. Peptide Interactions with Lipid Bilayers Using Molecular Dynamics Simulations

Many biological processes are associated with the peptide interaction with lipid membranes. Peptides exert their action after the adsorbtion or insertion in the bilayer by different mechanisms [192]. They could promote lysing or permeabilizing the membrane, attaching to larger membrane proteins, or self-aggregate (forming pores for instance). The access to the molecular scale through simulations of peptide-membrane interactions could help to understand and design short peptides to be used as bioactive agents.

Nowadays, there are reliable models to investigate these problems through computer simulations, such as Molecular Dynamics technique [193]. Molecular dynamics simulation could help to study problems of many bodies based on classical mechanics. Within this approach, Newton’s equations are solved numerically [194]. The main advantage is the realistic simulation of materials through the simplification by potentials with analytical form. Due to the development of fast and efficient methods for treating long-range electrostatic interactions, significant improvement in computer hardware, algorithms, and reliable force fields.

Here we focus on two molecular description scales: Atomistic and coarse grain (CG). The main difference between them is that in an atomistic scale all the atoms are represented, whereas in a CG scale, atoms are grouped in beads [195]. In this way, CG models allow the reduction of the degree of freedom and the integration of Newton equations in a higher time step, due to the elimination of high-frequency vibration modes [196]. Figure 7 shows illustrative snapshots of all atom representation of a lipid bilayer and a coarse grain representation of a liposome, both in the presence of short cationic peptides.

What we can get from simulations of peptide interaction with lipid bilayers?

Using MD simulation at the atomistic and CG levels could bring very valuable information on the short peptide-membrane interaction. We mention here the issues and consider that this type of methodology has its contributions and challenges.

#### 6.3.1. Peptide Affinity Dependency on Membrane Composition

The partition and insertion of single peptides in membrane of different composition have been broadly investigated using unbiased MD simulations at both description levels, with very successful results [197]. Within this technique, it is possible to access to the molecular details and identify specific interactions (hydrogen bond, salt bridge, cation π), peptide conformation, as well as bilayers properties associated with the interaction (induced defects with a variety of morphologies, stiffness, fluidity, etc.).

#### 6.3.2. Free Energy Calculations through a Peptide Reaction Path

Important information could be obtained by calculating, for instance, the free energy through a given path for the peptide translocation along the bilayer. For bilayers, the natural path is the z direction (normal to the bilayer). The use of enhancement of the sampling of configurational space is well established where we can highlight that several methods such as umbrella sampling [198], adaptive biasing force (ABF) method [199], the Wang-Landau algorithm [200], steer molecular dynamics [201], and metadynamics have been proposed [202].

#### 6.3.3. Cooperative Effects

Among the proposed mechanisms of lithic action of peptides in membranes is the formation of pores. The study of cooperative effects, such as a pore formation, is not an easy task. It is hard to access with enhanced MD techniques or applying an electric field. Combining CG [203] and atomistic [204] level MD simulations could be used to broadly sample the phase space. Thus, different initial conditions could help to explore this space. For instance, pre-assembled pores embbebed in lipid bilayers could bring information on their stability dependence with the bilayer lipid composition [204].

## 7. Peptide-Based Therapies

### 7.1. Monocyclic, Bicyclic and Tricyclic Cell-Penetrating Peptides as Molecular Transporters

CPPs have become a subject of major interest [205,206] for the intracellular delivery of therapeutic agents because of the limitations associated with linear CPPs, such as endosomal entrapment, toxicity, poor cell specificity [207], poor stability and degradation [208], and suboptimal cell penetration. Several cyclic CPPs possess enhanced cell-penetrating ability and improved physicochemical properties and proteolytic stability. Some cyclic CPPs exhibit endosomal-independent uptake [209]. A few cyclic CPPs have been reported to have a nuclear-targeting property [209]. We have summarized below some of the monocyclic, bicyclic, and tricyclic CPPs containing arginine and other amino acids (Figure 8). We have reported the application of cyclic CPPs containing alternatively positively charged arginine and hydrophobic tryptophan residues, [WR]_4_ and [WR]_5_, as drug delivery tools and nuclear targeting tools in 2011 [209]. Several other monocyclic CPPs were engineered based on this template and were shown to be efficient molecular transporters for enhancing the efficacy of existing chemotherapeutic, antiviral, and antibacterial agents [210,211,212,213,214,215,216,217,218,219]. The monocyclic CPPs containing tryptophan and arginine residues were also used to conjugate with potential therapeutic agents. For instance, monocyclic peptides were conjugated with doxorubicin, paclitaxel, and camptothecin [216,220] demonstrating localization of the drug moiety in the nucleus in case of doxorubicin. Adding to the previous work, we also demonstrated that several monocyclic peptides containing cysteine and arginine residues, such as cyclic [CR]_4_, significantly enhanced the cellular uptake of a cell-impermeable phosphopeptide (F’)-Gly-(pTyr)-Glu-Glu-Ile (F’-GpYEEI) in the presence of endocytic inhibitors and sodium azide in the lymphoblastic leukemia cell line (CCRF-CEM) [218]. Furthermore, tryptophan and histidine-containing cyclic decapeptides [WH]_5_ demonstrated an increase of the intracellular delivery of a cell-impermeable phosphopeptide and an anti-HIV drug, emtricitabine [217]. In another effort, monocyclic [HR]_4_ peptides were used as a molecular transporter and were able to double the permeability of F’-GpYEEI in human ovarian adenocarcinoma cells (SK-OV-3) cells [214]. Two bicyclic peptides containing tryptophan and arginine residues, namely [W_5_G]-(triazole)-[KR_5_] and FAM-[W_5_E]-(β-Ala)-[KR_5_], significantly enhanced the cellular delivery of cell-impermeable F´-GpYEEI in SK-OV-3. The confocal microscopy exhibited that the peptides were localized in the nucleus and cytosol [219]. Recently, we designed a tricyclic peptide [WR]_4_-[WR]_4_-WR]_4_ containing alternate tryptophan and arginine in each ring that improved the cellular uptake of F’-GpYEEI and fluorescently labeled anti-HIV drugs, lamivudine (3TC), emtricitabine (FTC), and siRNA in the breast cancer cell line MDA-MB-231 [221]. A combination of tryptophan and arginine residues in the design provided a diverse class of cyclic CPPs with differential cellular delivery properties. These data revealed the potential of monocyclic, bicyclic, and tricyclic CPPs as molecular transporters and provided insights into the design of the next generation of peptides as drug delivery tools.

### 7.2. Short Peptides in Gene Delivery

A variety of oligopeptides have been proposed to overcome the extracellular and intracellular barriers in gene delivery. Peptide sequences can be incorporated in a complex gene delivery system (GDS) or used alone as a single carrier. Short peptides can be combined as constructs such as peptide dendrimers and potentially provide more than one feature in a single system. Peptides used in gene delivery can be categorized as (1) targeting peptides, (2) cell penetrating peptides, (3) endosome disruptive peptides, and (4) nuclear localization peptides.

#### 7.2.1. Targeting Peptides in GDSs

Small peptide sequences have extensive applications as targeting moieties on the surface of GDSs. Conventional IgG antibodies (~150 kDa) are difficult and expensive to produce and introduce a high chance of immunogenicity. In contrast, small targeting moieties such as short peptides provide lower chance of immunogenicity due to lack of Fc region, increased ligand multivalency, and easy production by peptide synthesis methods [222,223,224]. Techniques such as phage display provides a means for identifying specific peptide sequences able to target selective tissues or cells [225,226]. A variety of peptides can be used either to target cell surface markers or subcellular elements in vivo. One of the most investigated target receptors on the cells, especially for cancer gene therapy are integrins, which are a class of cell adhesion molecules consisting of an α and a β subunit. Targeting moieties against α_v_β_3_ integrin, highly expressed on activated endothelial cells, contain a conserved arginine-glycine-aspartic acid (RGD) motif [227,228,229,230,231]. To achieve enhanced biological and pharmacokinetic properties, researchers have designed different RGD-containing peptide analogs and clustered RGDs [232,233]. Targeting ligands against other cell surface markers such as growth factor receptors (GFRs), transferrin receptors (TFRs), low density lipoprotein receptors (LRPs), acetylcholine receptors (AchRs), leptin receptors (LRs), and insulin receptors have been also investigated in targeted GDSs [234]. An important design factor in decorating GDSs with targeting peptides is accessibility of targeting moiety to interact with the target molecules effectively. Hindrance of the short peptide inside the GDS with loose interactions such as electrostatic interactions could lead to reduced targeting efficiency [235]. Other than cell surface, oligopeptides can be used to target the subcellular compartments such as endoplasmic reticulum, mitochondria, or Golgi apparatus. General nanoparticle targeting approaches to intercellular pathway/compartments can be found elsewhere [236,237,238,239].

#### 7.2.2. Cell Penetrating Peptides in GDSs

Cell penetrating peptides (CPPs) are capable of translocating cargo therapeutics across the plasma membrane [240]. Physicochemical nature of CPPs can be cationic, amphipathic, or hydrophobic. Cationic CPPs are generally composed of short sequences of arginine or lysine. Some well-known examples are truncated Tat peptide (GRKKRRQRRRPPQ) [241], penetratin (RQIKIWFQNRRMK-WKK) [242], transportan 10 (AGYLLGKINLKALAALAKKIL) [243], and KALA (WEAK-LAKALAKALAKHLAKALAKALKACEA) [244]. Tat is a peptide fragment from residues 48 to 60 of the original transcription activating factor of human immunodeficiency virus (HIV-1) [241] and penetratin is obtained from a Drosophila homeodomain protein [242]. Tat-modified GDSs have shown enhanced cargo transport across several biological membranes such as cellular, endosomal, and nuclear membranes [245,246,247,248]. For peptides such as penetratin, the non-electrostatic interaction to the non-polar parts of plasma membrane is also of great importance [249,250,251]. The number of cationic residues, spacing between them, and inclusion of non-peptide elements such as hydrophobic lipid moieties were discussed as important factors in designing optimal synthetic CPPs [252,253,254,255,256].

#### 7.2.3. Endosome-Disruptive Peptides in GDSs

Endosome-disruptive peptides destabilize the endosomal membrane leading to cargo release into the cytosol. Some provide membrane fusogenic features either pH-dependent or pH-independent. Sequences derived from the N-terminus of the influenza virus hemagglutinin HA-2 are hemolytic at pH 4, but not at pH 7 [257,258]. Protonation of the peptide acidic residues under acidic condition of endosomes would induce conformational changes leading to membrane binding and destabilization [258,259]. On the other hand, the hemolytic activity of melittin sequence is not pH-dependent [260,261]. MT20, a short peptide consisting of the first 20 residues of melittin, was designed to increase hemolytic activity at acidic pH and lower it at neutral pH [262]. Histidine-rich peptides are considered as another effective endosomolytic elements in GDSs. The imidazole ring in Histidine structure acts as a weak base that can be protonated at pH 6 of endosomal compartment [263,264]. The pH-dependent protonation of histidine and histidine-containing peptides result in pH-buffering as well as fusogenic capabilities linked with membrane disruptive effects [265,266,267,268]. Branched histidine-rich sequences have provided high gene and siRNA transfection efficiencies [269,270]. Additional improvements were also observed in cases in which polymeric gene carriers were modified with imidazole- or histidine-rich sequences [265,271,272,273]. Amphipathic sequences such as the peptide, which consists of a tandem repeat of glutamic acid–alanine–leucine–alanine (GALA), was also employed in GDSs to provide pH-sensitive fusogenic feature in acidic pH [274,275].

#### 7.2.4. Nuclear Localization Peptides in GDSs

Nuclear localization signals (NLSs) are reported to transfer cargos such as nucleic acids through the nuclear pore complex (NPC) of non-dividing or slowly dividing cells. They are believed to act as adaptors between the cargo and the importin-dependent nuclear transfer machinery. Short NLSs such as monopartite SV40 large T antigen with the sequence of PKKKRKV [274], or bipartite nucleoplasmin with the sequence KRPAATKK-AGQAKKKK [276], have been reported to provide enhanced DNA transfection efficiency [277]. Peptides derived from H1 histones, protamines, ribonucleoprotein A1, and high motility group (HMG) proteins could also direct DNA to the nucleus [278,279]. NLSs have been extensively used to modify GDSs for improved transfection capability [280,281]. Due to the cationic nature of most of these peptides, some believe that the enhanced gene delivery efficiency observed with them, may be related to factors other than effective nuclear recognition [282], including better polyplex morphology and DNA binding strength.

### 7.3. Taking Peptide Aptamers to a New Level

Aptamers are molecules whose structures have been conformed to adapt to a target ligand in such a way as to optimise binding interactions between the aptamer and that ligand. *Aptus* in Latin means attached, adjusted. They can be constructed from a range of diverse chemical species and while nucleic acids are some of the first templates to have been employed for this approach, peptide aptamers are attracting increasing attention because of their biocompatibility, wide variety of physical characteristics, ability to introduce modifications using standard chemical techniques, and ease of large-scale manufacture. In this review, the different methods employed for designing peptide aptamers are discussed, together with new methodologies that can extend yet further their huge potential. Peptide aptamers can be broadly defined as peptides that have been adapted to interact with a specific target, usually employing a methodology that is able to select out high binders from a combinatorial library. Like nucleotide aptamers, the methods used to date to generate peptide aptamers have relied on genetic amplification mechanisms. The archetype of this approach is phage display [283]. Starting with a library of randomly generated amino acid sequences spliced into large scaffold proteins, in this case coat proteins of a bacteriophage, those phages that bind to a target can be separated from non-binders and those grown up in host cells to harvest those sequences with the strongest interaction. The power of the technique is that every individual sequence has, physically associated with it, the nucleic acid strand that codes for it, so that separating out the binding peptide separates out the coding nucleic acid at the same time. The same principle applies to all other methodologies employed so far.

Instead of being expressed on the phage surface, the library can be expressed on the surface of the bacteria growing the phage [284], or can be simplified to a cell-free presentation system where modifications made to the coding RNA (removal of the stop codon) result in a ribosomal complex in which mRNA, ribosome, and nascent protein remain physically attached to each other [285]. Extensions of this approach give rise to more stable protein-RNA conjugates in which the ribosome departs the complex after chemical ligation transfers the RNA directly to the protein [286].

At the same time, the technique has also progressed in the other direction, so that the aptamer target binding interaction actually takes place inside the cell that generates the aptamer [287], allowing for a more diverse set of read-outs than simply binding interactions.

Many different scaffold proteins are employed, chosen for their stability and their ability to accommodate stretches of foreign sequences while still folding into a single, unequivocal tertiary configuration. Certain scaffolds are more appropriate for hosting binding epitopes in the form of clefts (e.g., “affibodies” [287]), while others present sequences in loops (e.g., “knottins” [288] or “atrimers” [289], which can be modified subsequently by chemical means to increase their rigidity still further [290]. Aptamers have been created to an enormous range of targets and a number are now on the market, either as therapeutics [291] or diagnostic agents [292]. Their appearance is bound to increase rapidly, as they are highly biocompatible, can avoid being targets of the immune system, and can be manufactured by standard means.

As drugs go, however, they are still quite large, in order for the scaffold to fulfil the requirement for ease and reproducibility of folding. The composition of the structures obtained is of necessity confined to amino acids, (although a limited range of non-natural amino acids can be incorporated), and the initial discovery process, particularly for the cell-free systems, requires prior knowledge of the target structure to which the aptamer must bind.

Recently, however, a combinatorial amino acid structuring technology has been described that overcomes these limitations and that can take development of peptide aptamers to the next stage. In this technique (termed Mozaic), combinatorial libraries are produced by presenting mixtures of amino acids in a close-packed 2D mosaic array on the surface of nanoparticles [293]. The nanoparticles are actually micelles and the amino acids are the head-groups of the amphiphiles used to form the micelles, which form spontaneously when the amphiphiles are dispersed in aqueous media. Starting with a pool of the 20 amino acids (supplemented if desired by sugars, steroids, heterocyclics, and other non-natural amino acid residues) a suspension of micelles can be prepared that contains a predetermined selection from the starting pool, all mixed together in all possible random configurations on the surface of those micelles. A whole library of micelles can be prepared in this way, each with a slightly different combination of amino acid monomers.

Each member of the library can then be tested in a bioassay to determine which combination of amino acids is most effective at eliciting a certain desired behaviour. The micelles are non-toxic and biocompatible, so are amenable to use in assays involving cell culture. Because large amounts of the building blocks are readily synthesisable, micelles can be produced in large enough quantities that they can bring about changes in the behaviour of cells, rather than simply binding to them. Such changes can include cell differentiation, cytokine secretion, or induction of apoptosis. The process is iterative, so that after one amphiphile combination has been identified as positive, modifications can be made (e.g., reducing the number of components, or substituting them for others) to improve the response still further.

Because the read-out is a change in cell behaviour, results can be obtained even in the absence of knowledge as to what receptor needs to be targeted to achieve a desired effect and indeed, the technique can be employed to discover new, hitherto unknown receptors on cells whose stimulation can bring about the behaviour sought.

Under certain circumstances it is possible to test the activity of the micelles in in vivo models for disease. Normally, however, because of the labile nature of micelles, it is not possible to employ them directly as therapeutic agents, and their conversion to more stable peptide aptamers, particularly planar ring structures mimicking the planar surface of the micelle, gives rise to molecules that can change cell behaviour at very low concentrations [294]. Some of the most successful are cyclic peptides as small as 10 amino acids (Figure 9), stabilised by cross-ring hydrogen bonds. Their potential in vivo has been demonstrated in experimental models for rheumatoid arthritis [294].

### 7.4. Peptide-Based Vaccines

Short peptides play crucial roles in the immune system and are responsible for the transmission of most of the immunological information [295]. The immune system does not recognize an antigen per se, instead recognizing the B-cell, CD4+ (T-helper), and CD8+ (cytotoxic T lymphocyte, CTL) epitopes. B-cell epitopes (5–20 amino acids in length) are conformational (e.g., helical) and, as the antibody binding site, they are critical in the generation of humoral, antibody-based, immune responses. CD8+ epitopes (9–11 amino acids) are non-conformational and are recognized by MHC-I cell receptors. They trigger cellular immune responses and activate CTLs. CD4+ epitopes (typically 12–16 amino acids) are recognized by MHC-II receptors in antigen presenting cells (APCs) and are responsible for “help”; they further activate both humoral and cellular immune responses.

Given this understanding, the most minimalistic vaccines can bear just those epitopes and still induce the desired immune responses. These vaccines, named peptide-based vaccines, have several advantages over classical vaccines (which are based on whole pathogens or their large fragments): (a) They induce precisely engineered epitope-specific immune responses; (b) they do not cause allergic, autoimmune, or excessive inflammatory responses; (c) they provide direct immune responses to non-immune dominant fragments of antigens; (d) they can be produced through chemical methods with high purity and reproducibility and can be precisely characterized in the same manner as small molecule-based drugs; (e) biological contamination is avoided due to chemical synthesis; and (f) they are more stable than whole pathogens or proteins and usually do not require “cold chain” storage or transport conditions [296,297].

However, peptides alone are poor immunogens and require immune stimulants (adjuvants) [298]. Many approaches have been investigated to stimulate and deliver peptide-based vaccines without the need for classical adjuvants (e.g., Alum) [299]. These delivery systems/adjuvants have been designed to: (a) Improve peptide delivery to APCs; (b) protect peptides against enzymatic degradation; (c) maintain the desired conformation (crucial for antibody-based immune responses); (d) create a depot effect at the injection site and control the release of antigen (mimicking natural local infection); (e) deliver vaccines to specific tissues (e.g., lymph nodes, spleen); (f) allow cross-presentation (mimicking viral infection); and/or (g) carry multiple epitopes covering different pathogen serotypes, stages in the pathogen life cycle, or even epitopes derived from different pathogens [300,301,302].

Interestingly, short peptides can also serve as adjuvants and delivery systems. For example, upon conjugation with a peptide antigen, Q11 peptide (QQKFQFQFEQQ) self-assembled into β-sheet fibrils. This strategy has been used to deliver vaccine candidates against malaria [303], tuberculosis [304], and group A streptococcus (GAS) [305]. Polyleucine (L15), when conjugated to B-cell epitope, self-assembled into nanoparticles. These particles induced the production of high antibody titers and protected model animals against GAS [306] and hookworm [307] infections. Polyglutamic and aspartic acids (E10 and D10) formed self-adjuvanting polyelectrolyte-based complexes when mixed with peptide antigen and trimethyl chitosan (TMC), while a simple mixture of antigen with TMC did not induce strong immune responses [308,309,310]. Cell-penetrating peptides have been used for the delivery of a variety of vaccines, mostly targeting cellular immune responses [311]. For example, muramyl dipeptide (N-acetylmuramic acid modified *L*-alanine-*D*-isoglutamine) was approved as an adjuvant for a Leishmania animal vaccine [312]. Peptide-based vaccine approaches are relatively new and no peptide-based vaccines are available on the market. However, a huge number of peptide-based vaccines—targeting practically all possible infectious diseases, including acquired immune deficiency syndrome (AIDS), malaria, and coronavirus disease 2019 (COVID-19)—are under development. Covaxx anti-COVID-19 peptide vaccine entered clinical trials and its UB-612 vaccine will be tested in phase II/III clinical trials in Brazil [313]. The number of peptide-based vaccines in clinical trials has grown significantly over the last decade; several of these, particularly those against cancers, have reached phase III [314,315].

### 7.5. The Role of Short Peptides in Neurodegenerative Therapy

Short peptides are a vital and promising tool in the study of neurodegenerative disorders (Alzheimer’s, Parkinson’s and Huntington’s diseases, amyotrophic lateral sclerosis, prion diseases). They can be used in either diagnosis or further treatment. As an example, short peptide-based inhibitors of amyloid β aggregation in Alzheimer’s disease with reduced cytotoxicity are good drug candidates [32,33]. Notably, very short peptides, like diphenylalanine and related compounds such as *tert*-butoxycarbonyl derivative (Table 2 [126]), are the core recognition motifs of β-amyloid polypeptides [98,126]. Fibrillar dynamic self-assembly behaviour of short peptides, leading to amyloid-like supramolecular structures, is a key insight into the nucleation, oligomerization, and the physical properties of amyloid fibrils [126].

Theses very short peptides as well as cell-penetrating peptides are of relevance to the highly efficient drug delivery systems (to target cells and subcellular organelles) and the prevention of amyloid β aggreagation [316]. Interesting and promising aspects of delivery strategy are peptide-based gel matrices to drug encapsulation [126]. Peptide-based therapeutic drugs induce the neuronal growth, modulate neurogenesis [317], and finally improve spatial memory. Their diversity prompts personalized therapies [40,318].

### 7.6. Immune Modulation Using Altered Peptide Ligands in Autoimmune Diseases

Altered peptide ligands (APL) refer to immunogenic peptides where the T cell receptor (TCR) contact sites have been manipulated, resulting in altered T cell responses. As such, 1–3 amino acid mutations have been designed for immune modulation in autoimmune diseases (Figure 10) [319]. Primary biliary cholangitis is a disease characterized by inflammatory destruction of small bile ducts in the liver leading to cirrhosis. Autoantibodies and T cells are associated with primary biliary cholangitis. The E2 region of pyruvate dehydrogenase complexes 159–167 (PDC-E2) binds to HLA-A2, and substitutions of alanine at position 5 of PDC-E2 peptide 159–167 leads to antagonism, by reducing peptide specific effector functions of the CD8+ T cells [320]. Likewise, in myasthenia gravis, an autoimmune disease regulated by CD4+ T-cells, which recognize the peptide epitopes p195–212 and p259–271 of the human acetylcholine receptor alpha-subunit, single amino acid mutations are able to inhibit myasthenia gravis autoimmune responses in mice. APL of these peptides up-regulate tumor necrosis factor-beta and down-regulate interferon (IFN)-gamma and IL-2 by the native peptide specific CD4+ T cells [321].

In addition, type-1 diabetes is characterized as inflammatory destruction of beta-islet cells within the pancreas by which autoantibodies, CD4+ T cells and CD8+ T cells, have been identified amongst other immune cells. Auto antigens have been identified, which include glutamic acid decarboxylase 65, non-specific islet cell auto-antigens, insulin, proinsulin, insulinoma antigen-2, imogen-38, protein-tyrosine phosphatase-2, zinc transporter-8, chromogranin A, pancreatic duodenal homeobox factor 1, and islet amyloid polypeptide [322]. The B chain of insulin epitope B_9–23_ is recognized by auto-reactive CD4+ T cells, which secrete high pro-inflammatory IFN-gamma. Mutating two TCR contact amino acids was able to induce secretion of anti-inflammatory IL-4, IL-5, and IL-10 [323]. Based on this data, the APL of B9–23 entered into human clinical trials in recent onset type-1 diabetic patients and although in phase 1 it was shown to increase anti-inflammatory T helper-2 (Th2) cells over pro-inflammatory Th1 cells, in a phase 2 study, it did not improve beta-cell function [324]. Similarly, APL from imogen-38 peptide epitope 55–70 was able to inhibit proliferation of CD4+ T cells and mannosylation was more effective compared to non-mannosylated APL [325].

Multiple sclerosis is an inflammatory disease involving immune cell infiltration and destruction of the myelin sheath. CD4+ T cells, CD8+ T cells, macrophages, auto-antibodies, Th17 cells, and others are involved in the pathophysiology of multiple sclerosis [326,327,328]. Numerous auto-reactive CD4+ T cell epitopes have been identified against myelin basic protein, proteolipid protein, and myelin oligodendrocyte glycoprotein, and APL of some of the epitopes have been evaluated. In particular, myelin basic protein epitope 83–99 and the shorter peptide 87–99 (MBP87–99) with one or two amino acid mutations are able to stimulate Th2 CD4+ T cells and antagonize Th1 CD4+ T cells [76,329,330,331]. Mannosylation of APL further enhances Th1 to Th2 diversion in mice and to human peripheral blood mononuclear cells from multiple sclerosis patients [332,333]. An APL of the longer peptide, MBP83–99, was tested in human clinical trials in patients with multiple sclerosis [320]. CD4+ T cells were stimulated which secreted high levels of IL-5 compared to high levels of IFN-gamma prior to treatment. These results indicate that APL immune modulation in patients with multiple sclerosis can stimulate Th2 responses, diverting reactivity away from the potentially harmful Th1 type [334]. In other studies, peptides from human papilloma virus containing part of the amino acid sequence (VHFFK) of MBP87–99 motif (VHFFKNIVTPRTP) induced experimental autoimmune encephalitis in mice, however, peptides from human papilloma virus type40 (VHFFR) and human papilloma virus type32 (VHFFH) prevented experimental autoimmune encephalitis [335]. Hence, microbial peptides, differing from the core motif of the self-MBP antigen, MBP87–99, may function as natural APL in the modulation of autoimmune disease.

Dysregulated immune cells, in particular CD4+ T cells, are amongst the cause and consequence of autoimmune diseases. Altering the fate of the auto-reactive CD4+ T cells by either switching off pathogenic cells, or altering their cytokine profile from pro- to anti-inflammatory is one mechanism of managing disease progression. The APL concept, which can influence T cell activation and polarization, has shown promise in animal models and in some human clinical trials as novel treatment modalities for auto immune diseases in the future.

### 7.7. Relevance of Short Peptides in Stem Cell Research

Stem cell therapies are part of regenerative medicine aimed at improving the quality of life through the fight with so far untreatable diseases. Short peptides are perfect candidates for the development of 2D and 3D stem cell-culture materials due to their excellent opportunities. First of all, peptides mimic the protein functions. They inter alia interact with DNA acting as regulatory factors. Oligopeptides imitate the extracellular matrix (the most important part of the stem cell niche), which influences the fate of the stem cell [336,337]. Short peptides play an important role in the transmission of bio-information, modulation of transcription, or restoration of genetically conditioned alterations being developed with age [317,338]. In other words, they have geroprotective properties. The self-assembling peptides in conjunction with stem cells are enable to regenerate damaged tissues [339]. They have relevance to the attenuation of the pathology of neurodegenerative diseases, heart failures, or arthritis [339]. Short peptides are involved in the regulation of proliferation and differentiation of stem cells into individual cell-types [340]. Good examples can be: Lys-Glu, Ala-Glu-Asp, Lys-Glu-Asp, Ala-Glu-Asp-Gly [317]. Modified peptides (e.g., by mentioned earlier Fmoc moiety) enhance differentiation activity. Peptide-based hydrogels are useful in controlling differentiation [341]. Very short peptides can penetrate the epidermis sending signals to cells. Peptides related to stem cells show great potential in the development of novel treatments, advance tissue regeneration, further rational design of the extracellular matrix-materials (as stem cell culture substrates), and organ engineering applications.

### 7.8. Short Peptide-Based Anti-Viral Agents against SARS-CoV-2

COVID-19 is an ongoing worldwide pandemic caused by SARS-CoV-2. So far, there are no specific antiviral drugs available for the treatment of this disease. However, a large number of small molecules displaying significant inhibitory activity against SARS-CoV-2 have been identified based on experimental and computational studies [342], among which many are short peptides and peptide-like compounds.

Generally, the peptide-based SARS-CoV-2 inhibitors target two proteins, namely membrane (M) and spike (S) proteins [343]. The former plays a crucial role in facilitating virus entry by mediating its interaction with the host cell receptor ACE2 and the latter (also called 3CLpro) is mainly in charge of the cleavage of viral polyproteins. Due to the high sequence similarity of different CoV 3CLpros, some known peptide-based HIV protease inhibitors have been used in clinical trials as the treatment of COVID-19, representative examples including lopinavir (1) and ritonavir (2) (Figure 11) [344]. Besides, some de nova designed 3CLpro inhibitors have also been identified based on structure-based rational design. For examples, Hilgenfeld and co-workers developed a series of α-ketoamide derivatives (e.g., Figure 11(3,4)), which was proved to be potent inhibitor of the SARS-CoV-2 3CLpro [345]; Liu, Jiang and co-workers discovered two peptidomimetic aldehydes (Figure 11(5,6)), which showed excellent in vitro activity against SARS-CoV-2 3CLpro [346]. These compounds showed good in vivo pharmacokinetic properties and safety, thus holding a great promise to evolve into anti-viral agent against SARS-CoV-2. In addition, a variety of peptide-based leading structures have been identified through virtual screening approaches, which may also serve as the starting point for the development of new therapeutics for COVID-19 [347,348].

The protein-protein interaction of the S protein of SARS-COV-2 and the host cell receptor ACE2 has also been viewed as an ideal target for the development of anti-viral agents against SARS-CoV-2. Particularly, the cryo-EM and co-crystal structures of the receptor-binding domain of SARS-CoV-2 with human ACE2 have been disclosed recently [349,350], which paves the way to develop new entry inhibitors against SARS-COV-2. For example, a 23-mer peptide sequence derived from human ACE2 was designed and synthesized by Pentelute and co-workers, which can specifically bind to SARS-CoV-2-RBD with low nanomolar affinity [351]. Besides, some potential ACE2-based peptide inhibitors have been identified through computational approach and their efficacy needs to be further validated in practice [352,353].

### 7.9. Antimicrobial Lactoferrin-Based Peptides as Anti-COVID-19

COVID-19 severity accrues due to its extremely high infection transmission rates [354]. Therefore, there is an urgent need for a novel, effective, and safe vaccine or drug to reduce the viral transmission rate and thus suppress the infection.

Different types of vaccines against SARS-CoV-2 infection have been under preparation [355]. However, some challenges related to vaccine administration safety have been of great concern. First, live or attenuated vaccines may recover their virulence leading to a high risk of disease recurrence to the vaccinee. Furthermore, transient immunosuppression may be induced resulting in the vaccinees’ susceptibility towards infections [356]. Second, DNA vaccines may cause a mutation risk upon integration with the host genome. Third, synthetic peptides are of low immunogenicity. Fourth, mRNA vaccines may need further quality and safety evaluation to enter the pharmaceutical market. Finally, yet most importantly, it was reported that vaccines produced from full-length spike (S) protein against SARS-CoV-2 might be involved in liver damage [355]. Some studies reported that sequenced strains of SARS-CoV-2 evolved into two subtypes (L and S), which showed great variation in geographical distribution, transmission ability, and disease severity. It is, then, speculated that the production of an efficient, safe, fully clinically approved SARS-CoV-2 vaccine may not take less than many years. Besides, the probability of the vaccines endorsement by the relevant authorities is considered low [355]. Many studies reported that Antimicrobial Peptides (AMPs), short sequence peptides polymer ranging from 10–100 amino acids, positively charged, amphiphilic, might be considered as a promising solution to combat harmful microorganisms [357]. Lactoferrin (LF), as one of the AMPs, is an iron-binding glycoprotein located at the mucosal layers of the human body. LF is considered as the first line of defense against microbial infection, which may have the potential to boost the innate immune response against COVID-19 [358]. Furthermore, LF is a natural, safe, and effective antiviral drug, which is naturally produced, in human and bovine milk, beside its anti-inflammatory and antitumoral properties. LF prevents the viral infection by blocking the host cell receptors or binding to the virus particles [358]. Moreover, some studies showed that the usage of LF as an adjuvant to vaccines might enhance their antiviral activity, beside being considered as a safe alternative to other used adjuvants [358]. Therefore, it may be recommended to use the LF as an immunity booster against SARS-CoV-2, rather than counting on the insufficiently tested vaccines, in order to spare the required time for full testing and clinical approval of the required vaccine. Finally, it may also be recommended to use LF in conjugation with the produced vaccine to enhance its anti-COVID-19 activity.

### 7.10. Peptides from Digestion of Proteins

Food proteins are long-chain polymers of amino acids, encrypted into which are peptides with potential health benefits, which may be used for the treatment and management of chronic and severe degenerative diseases such as hypertension, diabetes, obesity, cancer, and metabolic disorders [359]. Bioactive peptides (BAP) from food proteins have the amino acid structure and sequences similar to those that convey various signaling mechanisms or hormones in our body. They are small molecular weight peptides usually around 5 kDa. They have high tissue affinity, specificity, and efficiency to interact with receptors, enzymes, and other biomolecules in the body to confer health promoting effects [360,361]. Some of these BAPs may be released in the gut when proteins are degraded by the digestive enzymes such as pepsin, trypsin, chymotrypsin, and peptidases. However, controlled enzymatic digestion of food proteins in vitro would release some of these BAPs, which can be isolated and purified for therapeutic use.

The enzymatic production of bioactive peptides in vitro is greatly influenced by factors such as the pH, degree of hydrolysis (DH), enzymes used, enzyme/substrate ratio, temperature, hydrolysis time, and solvents used. It is well known that an abundance of hydrophobic amino acids, such as Gly, Val, Ile, and Ala, in the peptide sequences compared to the presence of other polar and charged amino acids, will contribute to the high bioactivity observed [361].

Bioactive peptides are naturally formed by the exogenous protease enzymes produced by microorganisms during fermentation. Some of the BAPs also have high antioxidant activities. Therefore, cultured milk products, fermented fruits and vegetables, and fermented meat and fish products are considered beneficial adjuncts in human diets.

### 7.11. Nutraceuticals

At present people ingest foods not just to cover their nutritional necessities; they also request healthy, natural, and convenient foods with biological activity. Interestingly, some plant proteins encrypt diverse peptides with beneficial effects on health. The most studied effects include anti-hypertensive, -cholesterolemic, -oxidant, -inflammatory, -cancer, -microbial, and immunomodulatory properties. Nowadays, different scientific areas have focused their research on the functional properties of foods and food products. Bioactive substances are known as components of foods that modulate metabolic processes, provide health benefits, and positive impact on the function of the body. A healthy diet is a key factor to prevent some diseases [362,363]. The intake of proteins is essential for maintaining a good health and it also provides bioactive peptides. Specific amino acids sequences encrypted in proteins in several foods have health effects, playing roles as fragment of the whole protein (epitopes responsible for interactions between proteins and antibodies) or being efficient after the release by proteolytic enzymes. Peptides with two or three amino acids pass easily through the gastrointestinal tract to the blood. Proteins are a source of bioactive di- and tri-peptides, among others, with certain biological activity [363,364,365]. Bioactive peptides have been known for several years and identified in plant and animal sources and their interest has increased in the last decades. They comprise positive health effects, i.e., on blood pressure and lipid metabolism, as well as analgesic, anti-thrombosis, anti-atherosclerotic and opioid agents. Some peptides have more than one activity. They are also useful to improve absorption of minerals [366,367,368]. Bioactive peptides are usually a product of the hydrolysis by gastrointestinal digestive enzymes (pepsin, trypsin, and chymotrypsin), or by in vitro producers with specific enzymes, temperature, or pH. They may contain hydrophobic amino acids in their sequences, a positive charge, and the resistance to digestion by proteases and peptidases and a proline C terminal. Small peptides with a dipeptide of proline-proline at their C terminal are more resistant to degradation by proteases and peptidases of the stomach, pancreas, or intestine. Large peptides may be active outside the intestinal epithelium. Recent studies of crop proteomic data revealed that at least 6000 proteins may harbour bioactive peptides [368,369].

### 7.12. Marine Peptides

Marine bioactive peptides of diverse bioactivities, encompassing anti-inflammatory, anticancer, and antioxidant activities, have been discovered from non-edible marine organisms and seafood processing by-products [370]. An interesting example is a hydrostatin-SN1 (DEQHLETELHTHLTSVLTANGFQ), an anti-inflammatory peptide identified from the venom gland of sea snake Hydrophis cyanocinctus [371], see Figure 12. In vivo anti-inflammatory effects of the peptide have been demonstrated in murine models of lipopolysaccharide (LPS)-induced acute lung injury [372] and in dextran sulfate sodium-induced acute colitis [371]. Hydrostatin-SN1 suppressed LPS-induction of pro-inflammatory cytokines, namely, tumor necrosis factor alpha (TNF-α), interleukin-6, and interleukin-1β, in mice. In vivo evidence and study on LPS-treated RAW 264.7 cells indicated the possibility of hydrostatin-SN1 exerting its effect by interfering with the extracellular-signal related kinase 1/2 and nuclear factor-κB (NF-κB) pathways [372]. In the murine colitis model, hydrostatin-SN1 exhibited its anti-inflammatory effect by binding to tumor necrosis factor receptor 1 (TNFR1), hence disrupting the interaction between TNFR1 and TNF-α. This, in turn, inhibited TNF-α-mediated activation of the NF-κB and mitogen-activated protein kinase proinflammatory pathways [371]. Hydrostatin-SN1 was proposed to be a promising candidate for the development of treatments for acute lung injury [164] and inflammatory bowel diseases [371]. Bioactive peptides were also reported from the giant barrel sponge, Xestospongia testudinaria [373] and rough leather coral, Sarcophyton glaucum [374] KENPVLSLVNGMF derived from X. testudinaria was dose-dependently cytotoxic to the human cervical cancer cell line (HeLa), being 3.8-fold stronger than anticancer drug 5-fluorouracil. By contrast, the peptide showed only a marginal 5% cytotoxicity to Hek293, a non-cancerous, human embryonic kidney cell line [374]. Likewise, AGAPGG, AERQ, and RDTQ identified from S. glaucum were more cytotoxic to HeLa cells than 5-fluorouracil, besides low toxicity to Hek293 cells [374]. Together, the aforementioned findings point to the potential of the four peptides as candidates for future development of anticancer drugs. Antioxidant peptides have been isolated from the by-products of the fish, mollusk, and crustacean processing, e.g., fish scales, fish skin, squid skin, and abalone viscera [375]. The ability of such peptides to dampen lipid oxidation and preserve seafood quality during processing and storage points to their potential application as preservatives [375]. Such peptides are also promising candidates in the development of therapeutics, adjunct therapeutics, or nutraceuticals against oxidative stress-related diseases or conditions [376]. Overall, marine peptides, whether derived from non-edible marine organisms or seafood processing by-products, may have potential applications in the discovery of peptide-based therapeutic agents and formulation of nutraceuticals.

### 7.13. Peptide-Based Cosmeceuticals

Biologically active short peptides are important cosmeceuticals, i.e., agents linking cosmetics and drugs. They deliver bio-activity in support of aesthetic effects [377,378]. The term “cosmeceutical” was coined in 1984 by Albert Kligman [379]. Peptide-based cosmeceuticals acting against both intrinsic and extrinsic aging and improving the health and appearance of skin are becoming increasingly popular. Bioactive peptides have either pharmaceutical or cosmetic value and open a new avenue in the field of gerocosmetology [380,381]. They are used for collagen stimulation, wound healing, “botox-like” wrinkle smoothing, as well as antioxidative, antimicrobial, and whitening effects [378]. Topical peptides are classified as carriers [copper tripeptide-1 (Cu-GHK), manganese tripeptide-1 (Mn-GHK), signal peptides [palmitoyl hexapeptide-12 (biopeptideELTM, palmitoyl pentapetide-4 (matrixyl), palmitoyl tripeptide-1 (biopeptide CLTM), palmitoyl tripeptide-5 (syn-coll), elaidyl-Lys-Phe-Lys-OH (lipospondin), hexapeptide-11 or pentamide-6, tripeptide-10 citrulline (decorinyl) and neurotransmiter inhibitors [acetyl hexapeptide-3 (argireline), pentapeptide-3 (vialox), pentapeptide-18 (leuphasyl), tripeptide-3 (syn-ake), acetyl octapeptide-1/-3 (sNAP-8) peptides [382]. Moreover, several natural peptides, such as carnosine, keratin, soybean, silk fibroin, and black rice peptides cannot be neglected [383]. These natural peptides as cosmeceutical ingredients fit perfectly into the rules of sustainability due to their high biodegadibility, low toxicity, moderate manufacturing costs, and convenient scale-up production ability [382]. More specifically, the United Nation defined 17 Sustainable Development Goals for the better future of the world, in relation to either people or the environment. In general, the main idea could be defined as “one global goal: good life for all”. One of these 17 aims is “good health and well-being” [384]. The forecast for the global cosmeceutical market predicts an increase by ~10% in the next years. Its significant share will be related to peptide-based products.

## 8. Conclusions and Future Outlook

Short peptides exhibit a remarkable array of biological functions, which may be used by innovative therapies in almost all branches of medicine. They are synthesized and investigated by research groups spread all over the world. The number of publications and patents in the subject has been growing enormously over the last years. This global review reflects this situation. It is written by scientists from all continents of the world who tried to unveil “fifty shades” of short peptides with the emphasis on biomedical, diagnostic, pharmaceutical, and cosmeceutical applications. In particular, peptides can play either a leading role as drugs or a supporting role in diagnosis, treatment, cell penetration, or targeting, and many more. Peptide-based vaccines are an expected breakthrough in cancer, microbial, or allergen immunotherapies. Natural and synthetic short peptides, including peptidomimetics, find numerous applications in nanotechnology and are thoroughly investigated by structural bio-informatics and supramolecular chemistry. Moreover, the development of comprehensive in silico techniques combined with efficient advanced synthetic methods facilitates the production of peptide based chemical species of almost unlimited applicabilities. 

To sum up, short peptides can be a secret of idealized smart therapies.

## Figures and Tables

**Figure 1 molecules-26-00430-f001:**
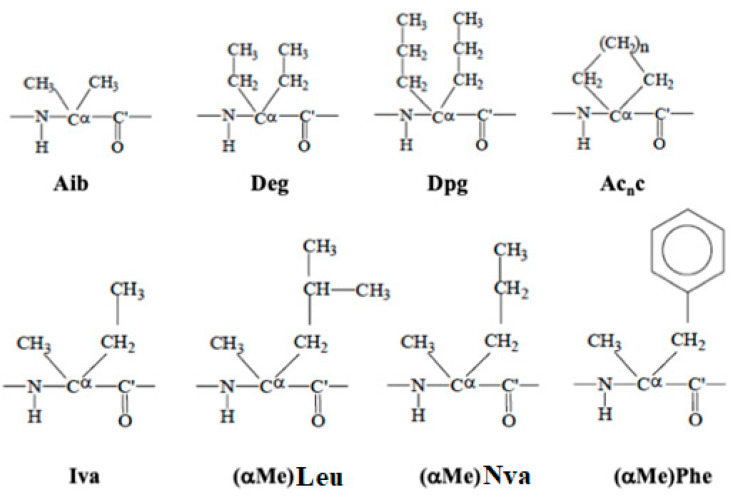
Schematic representation of some symmetrical C^α,α^-dialkylated glycine and chiral α-methylated residues.

**Figure 2 molecules-26-00430-f002:**
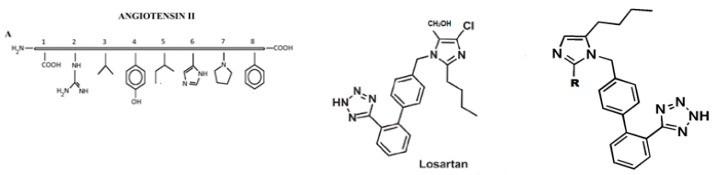
Angiotensin II (Asp-Arg-Val-Tyr-Ile-His-Pro-Phe), Losartan, V8 Losartan analogues (R = CH_2_OH, COOH). The angiotensin II scheme depicts the pharmacophoric groups of the eight amino acids of angiotensin II.

**Figure 3 molecules-26-00430-f003:**
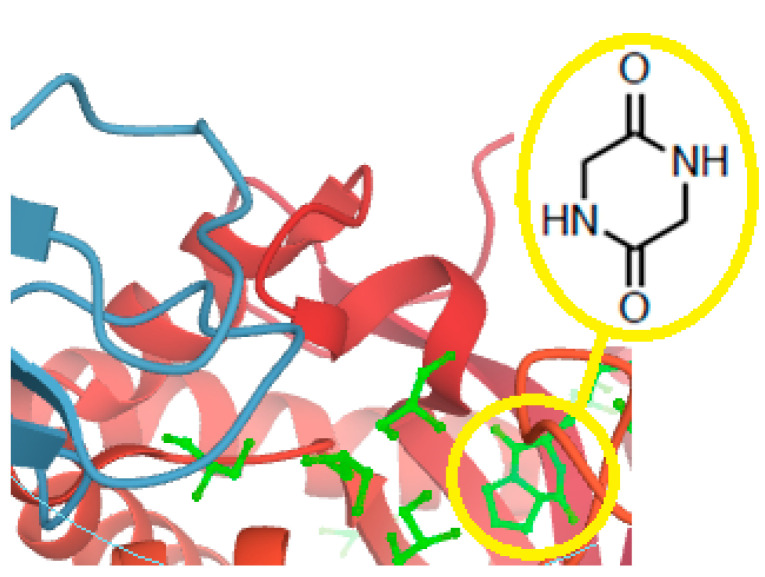
Bio-complex containg 2,5-DKP moiety [1O6I.pdb].

**Figure 4 molecules-26-00430-f004:**
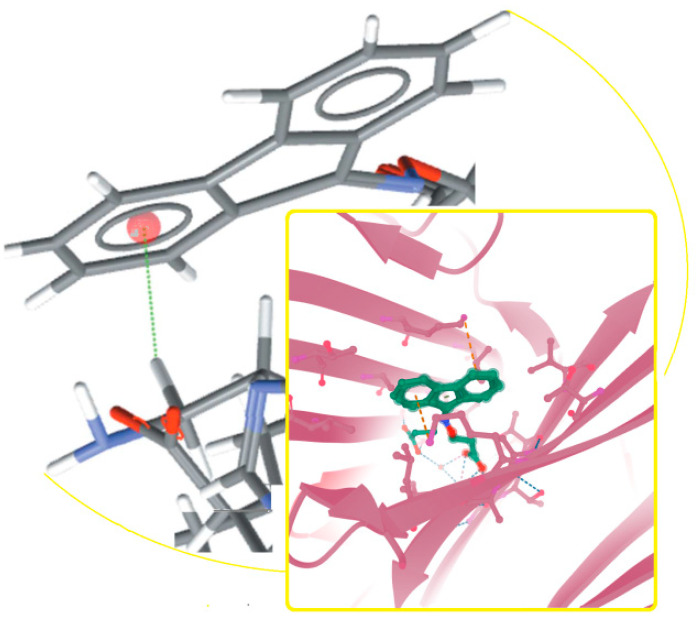
Bio-complex showing the C-H⋯π Fmoc interaction (PDB code 3gs4) [131].

**Figure 5 molecules-26-00430-f005:**
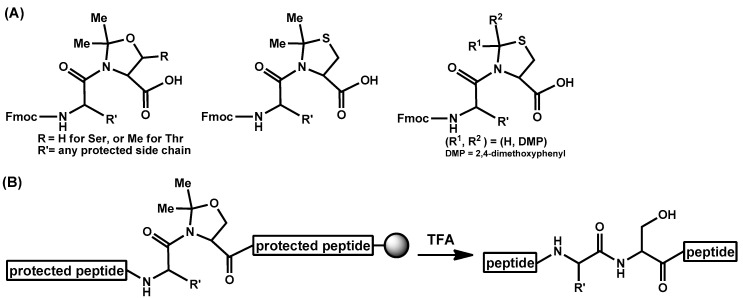
(**A**) Structures of Fmoc-protected ΨPro dipeptide units for Fmoc solid-phase synthesis (SPPS), (**B**) general scheme of ΨPro-aided SPPS.

**Figure 6 molecules-26-00430-f006:**
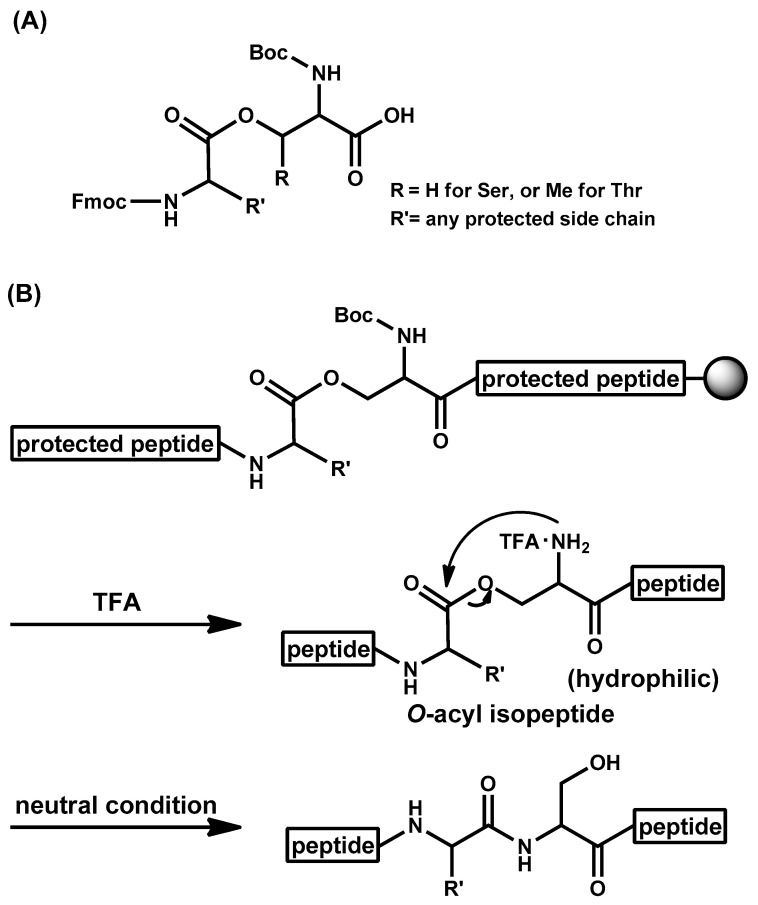
(**A**) Structure of the *O*-acyl isodipeptide units for Fmoc SPPS, (**B**) general scheme of the *O*-acyl isopeptide method.

**Figure 7 molecules-26-00430-f007:**
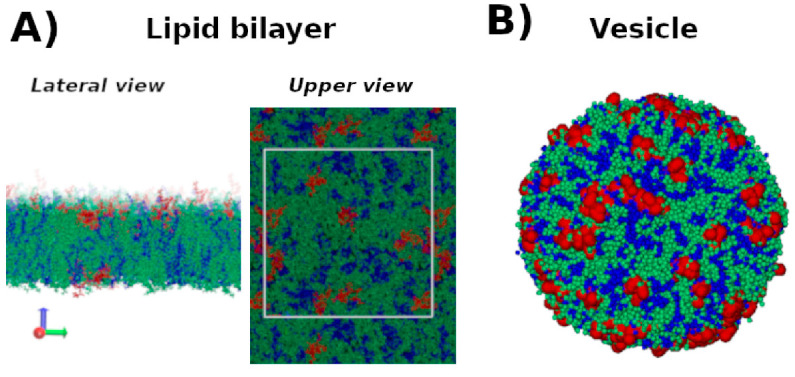
Representative snapshot of cationic peptides (red) adsorved into POPG (green)/POPE (blue) surfaces. Simulations were carried out using: (**A**) All atom model of Lipid bilayers and (**B**) Coarse grain model of vesicles. Water and ion sites were removed for visualization purposes.

**Figure 8 molecules-26-00430-f008:**
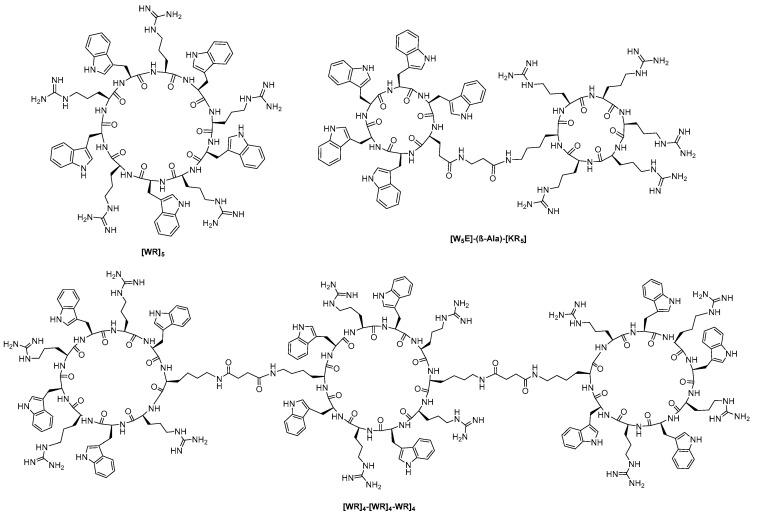
Monocyclic, bicyclilc, and tricyclic cell-penetrating peptides containing arginine and tryptophan residues as molecular transporters.

**Figure 9 molecules-26-00430-f009:**
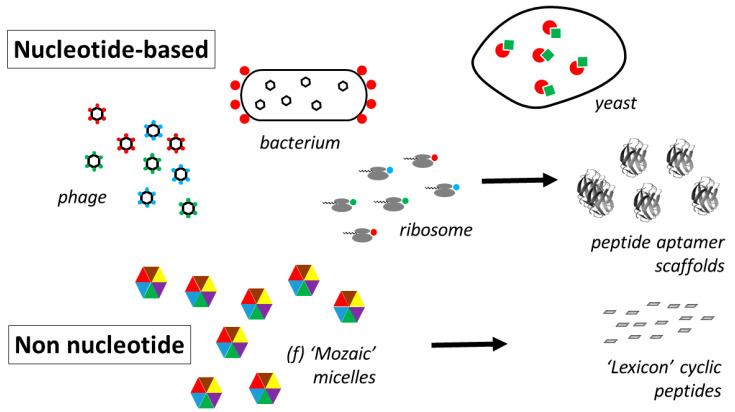
Presentation systems in peptide aptamer development.

**Figure 10 molecules-26-00430-f010:**
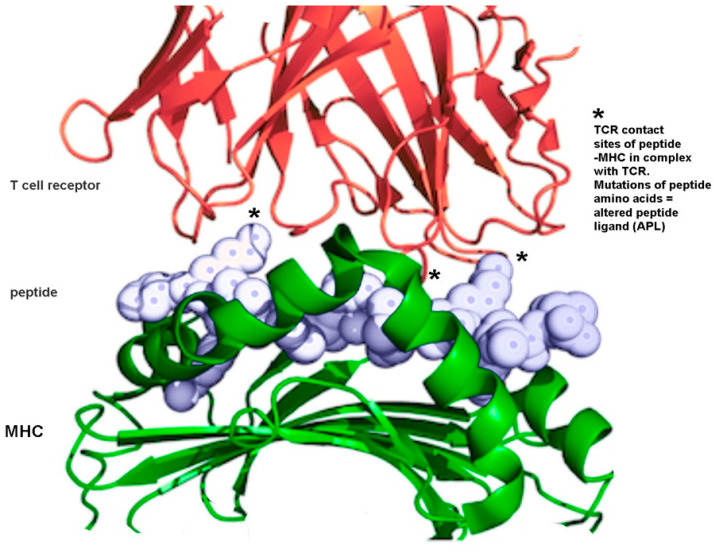
Major histocompatibility complex (MHC)-peptide-T cell receptor (TCR) trimolecular complex denotes amino acids from peptide TCR contact sites, where mutations result in APLs.

**Figure 11 molecules-26-00430-f011:**
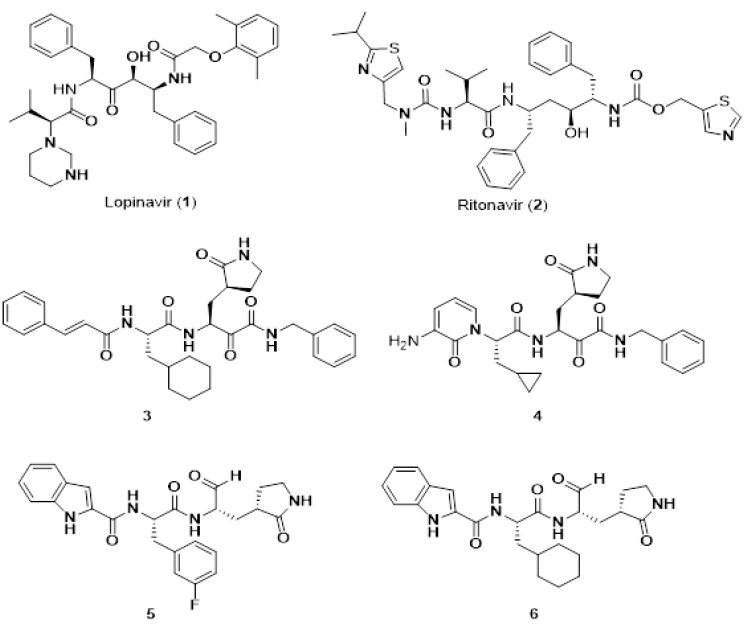
Selected peptide-based anti-viral agents against SARS-CoV-2.

**Figure 12 molecules-26-00430-f012:**
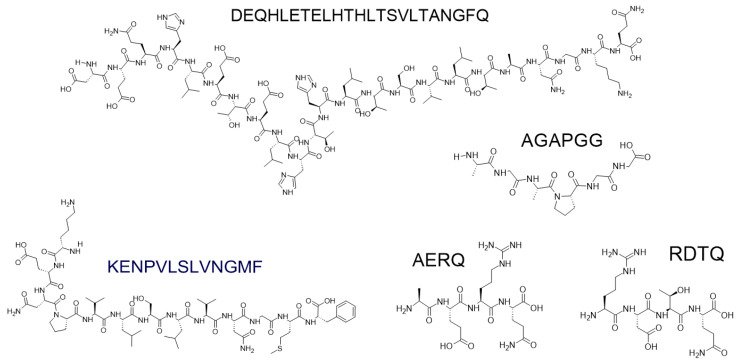
Anti-inflammatory (DEQHLETELHTHLTSVLTANGFQ) and cytotoxic (KENPVLSLVNGMF, AGAPGG, AERQ, and RDTQ) peptides derived from non-edible marine organisms.

**Table 1 molecules-26-00430-t001:** SWOT analysis of short peptides.

**Strengths**	**Weaknesses**
essential bio-molecules with a broad range of activities & functionalities in vivo	instability in vivo (easy degradation in plasma, protease sensitivity)
bio-chemical diversity, easy availability	short half-life
structural simplicity	low (oral) bioavailability
easy design & cost-effective synthesis with high purity	difficult membrane permeability in the case of greater peptides *
easy modification, scaling up	low binding affinity *
mechanical stability	high conformational freedom *
high: modularity, flexibility *, selectivity, target specificity, affinity *, absorbability, potency, tolerability, efficacy, safety, biocompatibility, biodegradibility	
low toxicity, antigenicity, immunogenicity	
easy recognition by bio-systems	
ability to penetrate the cell membranes (but only very short peptides) *, high brain penetration in systematical administration	
versatility as both targeting moieties and therapeutic agents	
specific interactions with various bio-systems	
predictable metabolism: degradation products are amino acids (non-toxic, natural entities used as nutrients or cellular building blocks)	
lack or fewer secondary off-targets (side) effects (peptides do not accumulate in kidney or liver)	
low unspecific binding to the structures other than the desired target, minimisation of drug-drug interactions, less accumulation in tissues (low risk of complications due to intermediate metabolites)	
**Opportunities**	**Threats**
development of peptide-based delivery systems: - cell-penetrating peptides - nano-cyclic peptide-based micceles, vesicles as gene or drug carriers - conjugations with non-peptidic motifs	oncogenicity of endogenous & synthetic peptides
supramolecular peptide-based biofunctional materials	immunogenicity (related to greater peptides)
formulations development (e.g., subcutaneous injections)various forms of using (drugs, vaccines, hormones, radioisotopes)	
development of the peptide-based safe & effective vaccines	
diveristy of well-ordered, robust, long-lived self-assembled nanostructures	
vital tool for neurodegenerative diseases studies & various applications in anticancer therapy	
peptoids or peptidomimetics	

* Bivalent property which may be either strength or weakness depending on particular species.

**Table 2 molecules-26-00430-t002:** Short peptides forming amyloid-like fibrils.

Name of Peptide	No. of Amino Acids
diphenylalanine	2
α,β-dehydrophenylalanine	2
Fmoc-diphenylalanine-konjac glucomannan	2
Ac-EFFAAE-NH2(AIP-1/2)	6
FFKLVFF	7
P11 (QQEFQWQFRQQ)	11

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
