# Peer review of "A Global Review on Short Peptides: Frontiers and Perspectives"

_molecules, 2021, doi:10.3390/molecules26020430_

Round 1

Reviewer 1 Report

In the present review, the authors propose a unique work related to the existing state-of-the-art of short peptides, considering many interesting aspects. Indeed, besides the description and classification of bioactive peptides, various aspects have been treated, such as the possible therapeutical developments, the functional properties of plants and marine biomolecules, also considering the latest evidences about SARS-CoV-2. Many information about their involvement in neurodegenerative therapies and autoimmune diseases have been proposed. In addition, authors considered the nanoengineering mechanisms adopted for developing peptide-based therapies, stem cells research, cosmeceuticals and the applications of cyclic peptides as molecular transporters. Interestingly, the authors also described the methods of synthesis, also highlighting possible difficulties, and in silico analyses or bioinformatics aspects.

Considering that many aspects have been well explained and the manuscript is well written, in my opinion this global review is valid for publication. Interesting is the idea of coordinating different researchers and experts from all continents of the world, developing a work that summarize many data and that emphasize the importance of the small peptides in future therapies.   

The only comment is related to the Figure 3. its quality might be improved, if possible. A better image would improve the already high quality of the manuscript. 

Author Response

Dear Reviewer,

Thank you so much for so perfect review.

We tried to correct quality of Figure 3 (now Fig. 10). Generally, Microsoft Word version result in poorer quality of diagrams.

Yours sincerely,

Authors.

Reviewer 2 Report

The manuscript entitled “A global review on short peptides: frontiers and perspectives” by Apostolopoulos et al., is written by the authors from the global continents and demonstrates the short peptide-based therapeutical developments. The manuscript summaries the discovery, definition and development of peptides from different kinds of aspects. It is useful for readers to understand the direction and recent peptide use. However, a major suggestion is recommended to help the manuscript for more completed. The authors should reorganize the Section 4 Bioactive peptides to extend the description for the peptides, such as Tumor homing peptides, Antioxidant peptides, Immune peptides, Peptide hormones, Amyloid peptides, Neuropeptides, Cell adhesion peptides, Antimicrobial peptides, and Antiviral short peptides, in Page 5. For example, Tumor homing peptides have cell penetrating features. However, the author did not describe what are the peptides with what is amino acid sequence, where is the homing peptides come from. In addition, how the homing peptides can be used to conjugate with drugs to lead to antitumor regression. Those descriptions in section 4 do not satisfy the readers how to understand the development of specific peptides. In addition, the descriptions of Section 4 are inconsistent for a similar format.

The following typos and suggestions should be revised, accordingly:

  1. Page 3 Line 117: “ofering” should be “offering” 
  2. Page 4 Line 137: “inidivisibly” should be “indivisibly” 
  3. age 4 Line 138: “nucleis ” should be “nucleic” 
  4. Page 6 Line 230: “neuropeptodes” should be “neuropeptides”
  5. Page 12 Line 465: “sever” should be “serve”
  6. Page 15 Figure 3: 1. The Ribbon diagram is so blurred. 2. Peptide in spheres should labeled with residue name in this case.
  7. Page 22 Line 848: “freeedom” should be “freedom”
  8. Page 26, Figure 7: “Asp-Arg-Val-Tyr-Ile-HisPro-Phe” should be “Asp-Arg-Val-Tyr-Ile-His-Pro-Phe”
  9. Page 26, Figure 7: In the picture of Angiotensin II, the sidechain of the first two residues (Asp and Arg) should be shown completely. 
  10. Page 32 Line 1155: the first PPI should present in full name.
  11. Page 34, Line 1223: “combaining” should be “combining”

Author Response

Dear Reviewer,

We significantly rebuilded Section 4 entitled Bioactive peptides. We tried reconcile the extremes in opinions both Reviewers (2 & 3). We removed some sub-titles and transfered this part (including „Peptides from digestion of proteins”, „Nutraceuticals”, „Marine peptides”). Moreover, we corrected text according the suggestions. In particular:

  • Page 3 Line 117: We changed “ofering” to “offering” .
  • Page 4 Line 137: We corrected “inidivisibly” to “indivisibly”.
  • Page 4 Line 138: We replaced “nucleis ” to “nucleic”.
  • Page 6 Line 230: We substituted “neuropeptodes” by “neuropeptides”.
  • Page 12 Line 465: We changed “sever” to “serve”.
  • Page 15 Figure 3: We tried to correct this Fig. (now Fig. 10, see earlier response to Reviewer 1).
  • Page 22 Line 848: We replaced “freeedom” by “freedom”.
  • Page 26, Figure 7: We corrected “Asp-Arg-Val-Tyr-Ile-HisPro-Phe” to “Asp-Arg-Val-Tyr-Ile-His-Pro-Phe”.
  • Page 32 Line 1155: We introduced full name of PPI into the text.

Page 34, Line 1223: We replaced “combaining” by “combining”.

Yours sincerely,

Authors

Reviewer 3 Report

The review entitled "A global review on short peptides: frontiers and perspectives" is a very extensive review that tries to embrace a huge list of aspects and applications as physiological role, therapeutic, identification, synthesis, in silico screening of short peptides. My main concern is the organization of the whole content. It is very confusing, repetitive in some points, and superficial in others. As the authors tried to discuss a variety of subjects they failed considerably in the organization of the text. In my opinion, they should restrict the approach to add some depth to the discussion. Also, to deliver a more organized review of the main literature in the field. I list some specific suggestions below. For the reasons stated, I do not recommend the publication of the review as it is. A considerably shortened and organized text would be a contribution to the field
My suggestions:
1. Abstract - The abstract needs to be considerably shortened. The abstract aims to present the reader with an overview of what he/she is going to read. Several pieces of information can be removed or moved to the Introduction. Decrease the number of adjectives, be concise.
2. Introduction - here the author can give a background up to the point they will start to review, shortly, concisely. The text presented is short but does not deliver what is expected from the introduction.
3. The expression "...peptides containing up to seven amino acids. However, it reflects no scientific worth" is wrong and contradictory. Several short peptides are scientifically/biologically/pharmaceutically relevant as the text will say later.
4. The subtitle "Bioactive peptides" is very broad as they can be physiological, in disease, as drug bioactive peptides, but the text that follows this subtitle is about "anticancer peptides". Then, several sub-sub titles (in italics) describe very shortly and superficially antimicrobial, antiviral, hormones, etc. So, what is a bioactive peptide? Is a hormone a bioactive peptide? Or do you mean therapeutically active? Very confusing and this text needs to be fully reorganized.
5. The other subtitle "Bioactive peptides from nature" probably means from food. Based on the text it seems the authors want to describe peptides originated from the digestion of proteins. My suggestion to improve the text is to change the titles. The same applies to "Nutraceutical and functional properties of bioactive peptides from plant foods".
6.In the subtitle "Peptide-based vaccines" all the paragraphs talking about adjuvants shall be shortened.
7. In the topic "Lactoferrin as anti-COVID-19" the paragraphs describing COVID-19 and vaccines should be cut. It is totally out of the context and it is repetitive.

8. In the topic "Short peptide-based anti-viral agents against SARS-CoV-2". Why choose to superficially describe the in silico screening of molecules for SARS-CoV-2 than describe the well-characterized interactions of Ritonavir in HIV-1 infected cells? The peptides investigated for SARS-CoV-2 could be mentioned but the interactions of ritonavir are very well described.
General comments
Some sequences of important peptides are not mentioned. A few figures of structures are present.
A more concise text will open space for relevant figures.

Author Response

Dear Reviewer,

 we send our answers:

*” My main concern is the organization of the whole content”, „As the authors tried to discuss a variety of subjects they failed considerably in the organization of the text”, „A shortened and organized text would be a contribution to the field

Answer: We significantly reorganized and rebuilded manuscript. As a consequence, order and numbers of the sub-sections are different now. Some parts were eliminated (e.g. Part 4 entitled Bioactive peptides), while other parts, such as: 7.5. Role of short peptides in neurodegenerative therapy and 7.7. Relevance of short peptides in stem cell research, were complemented by additional information.

*”Abstract - The abstract needs to be considerably shortened”. Several pieces of information can be removed or moved to the Introduction. Decrease the number of adjectives. Introduction - here the author can give a background up to the point they will start to review, shortly, concisely. The text presented is short but does not deliver what is expected from the introduction.  

Answer: Abstract has been shortened, while introduction changed. Some point were transfered from Abstract to Introduction. We eliminated number of adjectives.

*”The expression "...peptides containing up to seven amino acids. However, it reflects no scientific worth" is wrong and contradictory. Several short peptides are scientifically/biologically/pharmaceutically relevant as the text will say later”.

Answer: We agree. It could be misleading. We changed sentence to: “Ultra-short peptides were defined as a peptides containing up to seven amino acids”.

* ”The subtitle "Bioactive peptides" is very broad… this text needs to be fully reorganized”  „The other subtitle "Bioactive peptides from nature" probably means from food. Based on the text it seems the authors want to describe peptides originated from the digestion of proteins. My suggestion to improve the text is to change the titles. The same applies to "Nutraceutical and functional properties of bioactive peptides from plant foods".

Answer: Section was significantly rebuilded and shortened. We changed the titles. The parts entitled „Peptides from digestion of proteins”, „Nutraceuticals”, „Marine peptides” (new names) were transfered to the section „Peptide-based therapies”. Remaining sub-sections were eliminated.

*”In the subtitle "Peptide-based vaccines" all the paragraphs talking about adjuvants shall be shortened.”

Answer: Text related to adjuvants was shortened.

*”In the topic "Lactoferrin as anti-COVID-19" the paragraphs describing COVID-19 and vaccines should be cut. It is totally out of the context and it is repetitive”

Answer: Text was changed according to the above suggestion.

*”In the topic "Short peptide-based anti-viral agents against SARS-CoV-2". Why choose to superficially describe the in silico screening of molecules for SARS-CoV-2 than describe the well-characterized interactions of Ritonavir in HIV-1 infected cells? The peptides investigated for SARS-CoV-2 could be mentioned but the interactions of ritonavir are very well described.

Answer. In this section, our major focus is to introduce the peptide-derived compounds as potential anti-viral agents against SARS-CoV-2. Generally, there exist three approaches to access these compounds: repurposed known drugs, validated lead compounds based on rational design and promising leading structures from virtual screening. Ritonavir is a known anti-HIV drug, and has been tested against SARS-CoV-2. We have mentioned it in the manuscript as the representative example of repurposing known drugs. On the other hand, we think that “the well-characterized interactions of Ritonavir in HIV-1 infected cells" is not closely relevant to the focus of our subject (SARS-CoV-2), and thus rather should not be discussed in detail.

In Addition:

*Majority of the titles of sections were renamed. Order of sub-sections is completeley different.

*English was corrected.

*We corrected old Figures (1, 2, 3, 10, 11) and added new figure (4) and table (2).

*We eliminated repetitive points (e.g. in section Bioactive peptides and part concerning SARS-Cov2). Part related to diketopiperazines has been shortened, changed and included to the sub-section „Ultra-short peptides: less is more”. Generally, text and references (from 435 to 384) were considerably shortened.

Yours sincerely,

Authors

Round 2

Reviewer 3 Report

Authors significantly improved the manuscript shortening parts of it and reordering. I consider the paper can e accept in the current form.